# On heating of solar wind protons by the parametric decay of large amplitude Alfvén waves

Horia Comişel[1,2], Yasuhiro Nariyuki[3], Yasuhito Narita[4,5], and Uwe Motschmann[1,6]

[1]Institut für Theoretische Physik, Technische Universität Braunschweig, Mendelssohnstr. 3, D-38106 Braunschweig, Germany
[2]Institute for Space Sciences, Atomiştilor 409, P.O. Box MG-23, Bucharest-Măgurele, RO-077125, Romania
[3]Faculty of Human Development, University of Toyama, 3190, Gofuku, Toyama City, Toyama 930-8555, Japan
[4]Space Research Institute, Austrian Academy of Sciences, Schmiedlstr. 6, A-8042 Graz, Austria
[5]Institut für Geophysik und extraterrestrische Physik, Technische Universität Braunschweig, Mendelssohnstr. 3, D-38106 Braunschweig, Germany
[6]Deutsches Zentrum für Luft- und Raumfahrt, Institut für Planetenforschung, Rutherfordstr. 2, D-12489 Berlin, Germany

**Correspondence:** H. Comişel
(h.comisel@tu-braunschweig.de)

**Abstract.** By three-dimensional hybrid simulations, the proton heating is investigated starting from a monochromatic large-amplitude Alfvén wave with left-handed circularly polarization launched along the mean magnetic field in a low beta plasma. We find that the perpendicular scattering is efficient in 3 dimensions and the protons are heated by the obliquely-propagating waves. The thermal core proton population is heated in 3 dimensions as well in the longitudinal and parallel direction by the field-aligned and the obliquely-propagating sound waves out of the parametric decay. The astrophysical context is discussed.

**Keywords.** Space plasma physics (Wave-wave interactions, Wave-particle interactions, Numerical simulation studies).

## 1 Introduction

Early in situ measurements at 1AU from VELA satellite (Bame et al., 1975) reveal that the velocity distribution function of solar wind protons is broader in direction perpendicular to the mean magnetic field (hereafter the z-direction) than in parallel direction. This velocity anisotropy indicates a higher perpendicular temperature than the parallel one. Marsch et al. (1982) have found by using Helios 1 and Helios 2 data that such an anisotropic plasma heating occurs in high speed solar wind streams from 0.3 to 1.0 AU. This problem of anisotropic heating of ions in solar wind and solar corona is vast and has been discussed for a long time in space plasma physics (see e.g., Ofman, 2010). Theoretical models (e.g., Tu and Marsch, 1995) based on cyclotron resonant- or non-resonant- processes have been proposed to explain the anisotropic heating of solar corona and solar wind.

Marsch and Tu (2001) have shown for the first time the observational evidence for the occurrence of the pitch-angle scattering of solar wind protons, driven by resonance with ion cyclotron waves propagating away from the Sun. The perpendicular broadening of the sunward part of the measured distributions has been explained through the pitch-angle scattering of so-

lar wind protons resonantly interacting with the outward parallel-propagating Alfvén waves. In a later paper, Marsch and Bourouaine (2011) have shown that the antisunward part of the proton distribution functions can be similarly shaped by the proton diffusion by the oblique fast magnetosonic and Alfvén waves propagating away from the Sun. According to numerical simulation studies (e.g., Araneda et al., 2007), the field-aligned part describing the tail or the proton beam of the velocity distribution functions can originate in the parametric decay of the Alfvén waves, a process predicted by theories (see e.g., Sagdeev and Galeev, 1969) and supported by observations (see e.g., Spangler et al., 1997).

Parametric instabilities play an important role in the dissipation of the large amplitude Alfvén waves with parallel or quasi-parallel propagation with respect to the mean magnetic field and in plasma heating by means of the ion Landau damping mechanism. Parametric instabilities including decay, modulational, and beat instabilities, have been extensively analyzed by theoretical studies (see e.g., Viñas and Goldstein, 1991a, b) or numerical magnetohydrodynamics (MHD) (e.g., Ghosh et al., 1993; Ghosh and Goldstein, 1994; Ghosh et al., 1994; Del Zanna et al., 2001) and particle-in-cell or hybrid (e.g., Terasawa et al., 1986; Matteini et al., 2010a, b; Verscharen et al., 2012; Nariyuki et al., 2012; Gao et al., 2013; Nariyuki et al., 2014) simulations. In the MHD picture, the plasma heating by the Alfvén wave can occur through generation and steepening of **ion acoustic** waves. A shock wave is formed as a result of the wave steepening at a late (and nonlinear) saturation stage of the parametric decay. In the kinetic picture, hybrid simulations prove that the heating mechanism is completed by kinetic effects and a beam can be created in the ion distribution function due to the non-linear trapping of protons (see e.g., Araneda et al., 2008; Matteini et al., 2010a, b). The velocity beam formation is however restricted by the conditions of low-beta plasmas (see e.g., Matteini et al., 2010a).

Here we address the question, "Is the stochastic ion heating stronger in a 3-D parametric decay?" Our question is motivated by two preceding studies. First, Ghosh et al. (1993, 1994) and Ghosh and Goldstein (1994) discovered from the 2-D numerical MHD study that a parallel-propagating Alfvén wave collapses into obliquely-propagating daughter waves by the parametric decay. Second, more recently, Gao et al. (2013) confirm in the 2-D hybrid simulation that obliquely-propagating Alfvén waves are indeed excited by the field-aligned parametric decay, and propose a heating mechanism of the ambient plasma in a stochastic fashion. When the daughter Alfvén wave propagates obliquely to the mean magnetic field, the types of particle trajectories can be more diverse (see the illustration in Fig. 1). The finding and the assumed mechanism above by Ghosh et al. (1993, 1994) and Ghosh and Goldstein (1994), and Gao et al. (2013) are yet limited to a 2-D numerical setup. Obliquely-propagating waves are limited to a plane spanning parallel and perpendicular to the mean magnetic field in the 2-D setup, whereas the wavevectors can have a higher degree of freedom in the azimuthal directions around the mean magnetic field.

We perform a 3-D hybrid plasma simulation for the parametric decay, and track the time evolution of the proton distribution functions. We find that the stochastic heating (i.e., pitch-angle scattering) occurs more quickly and the ions are heated most strongly in the 3-D treatment. Our finding that the particles can be more quickly heated by the 3-D parametric decay can be tested by in situ measurements by the upcoming heliospheric missions such as Parker Solar Probe (Fox et al., 2016) and Solar Orbiter (Müller et al., 2013).

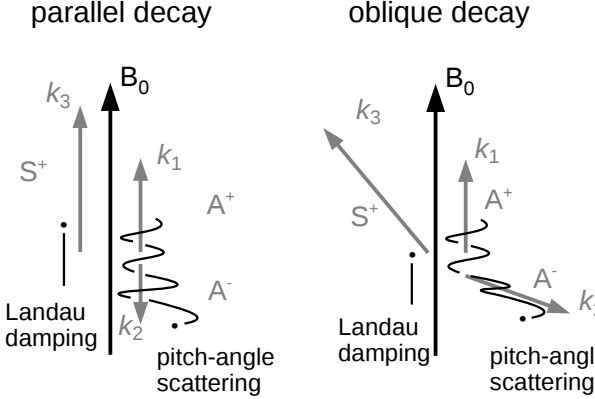

parallel decay  oblique decay

**Figure 1.** Parametric decay of a parallel-propagating Alfvén wave ('$A^+$' with the wavevector $k_1$) into a daughter Alfvén wave ('$A^-$' with $k_2$) and a sound wave ('$S^+$' with $k_3$) in the parallel decay scenario (left panel) and the oblique decay scenario (right panel). Particle trajectories are marked by solid lines in black.

## 2   Simulation setup and methodology

We perform hybrid simulations with the AIKEF hybrid code (Müller et al., 2011), conducted in a three-dimensional configuration: the size of the simulation box on each direction is L=288 $d_i$, the grid size is $1d_i$, and 1000 super-particles are used for each computational cell. Here, $d_i = V_A/\Omega_p$ is the ion inertial length while $V_A$ and $\Omega_p$ are the Alfvén velocity and ion frequency, 5  respectively.

The parametric decay modeled in the actual study is a three-wave process starting from a large-amplitude monochromatic Alfvén pump wave propagating parallel to the mean magnetic field $B_0$, a spectrum of electrostatic ion acoustic waves also at parallel propagation, and a spectrum of Alfvén daughter waves at anti-parallel propagation. The amplitude of the pump wave with left-handed circularly polarization is normalized to the value of the ambient magnetic field and has a value of 0.2 10  while its wavenumber and the resonant frequency are $k_0 \approx$ 0.218 $V_A/\Omega_p$ and $\omega_0 \approx 0.19\ \Omega_p$, respectively. The initial fluctuating magnetic field ($\boldsymbol{B}_\perp$) and bulk velocity ($\boldsymbol{u}_\perp$) satisfies the relation $\boldsymbol{u}_\perp = -k_0/\omega_0\boldsymbol{B}_\perp$. The resonant frequency $\omega_0$ is determined from the dispersion relation $k_0^2 = \omega_0^2/(1-\omega_0)$ for the left-handed waves, (see e.g., Terasawa et al., 1986). The seed amplitudes for the daughter sound waves are implicitly included by the simulation noise. A low value of $\beta$=0.01 is used for the plasma beta parameter for each species of particles which is relevant for the solar corona and inner heliosphere studies. **The used spatial** 15  **resolution for the field quantities (magnetic field, electric field, and velocity moments) is close to the ion inertial length and the proton gyroradius ($\rho_i \sim 0.1d_i$) or smaller spatial gradients cannot be resolved. The magnetic field within a numerical cell is overall homogeneous with the linear interpolations at the particle position between mesh points or due to the wave magnetic field. Thus, the perpendicular projection of the proton motion is nearly a circle and this circular**

gyration is resolved by about 100 time steps. Gradients become important over about 10 gyroradii and not just over one gyration.

We are warned that numerical heating could have some contribution in our simulations. Among various candidate mechanisms causing numerical heating one may specify: the numerical noise given by the statistical representation of the distribution functions, the rounding error or cutoff error when evaluating the differential operator, the absorption of the numerically-arising electric (possibly the electrostatic field) by the ions, and the random scattering due to the numerically fluctuating magnetic field (here the magnetic diffusion may be applicable). The numerical free energy occurring in the system can be converted in wave energy. This wave energy can be absorbed by particles and heating of the plasma. The heating effects described above can be compensated by using a suitable resistivity parameter, a smoothing procedure for the magnetic field, and numerical tests including various parameters. We have tested simulation runs with or without using pump wave by varying the number of particles per cell, time steps $\delta t$, and grid sizes to find out sufficient energy accuracy (within 5% for 500 elapsed ion-gyroperiods). Thus we conclude that the numerical heating does not play a significant role compared to the physical heating. The protons are treated in the hybrid scheme as particles while the electrons considered as a massless fluid. The value of $\beta$ parameter and of the pump wavenumber $k_0$ are selected such that the decay instability has growth rates larger than other parametric instabilities (e.g., beat instability and modulational instability expected for left-handed polarized waves) and can be safely evaluated by MHD theories. By using the analytic study discussed by Terasawa et al. (1986) in the two-fluid description of plasma, the growth rate of the decay instability has a maximum value of $\gamma_{th}$=0.0358 corresponding to a compressional wave excited at wavenumber $kV_A/\Omega_p$=0.385. More weaker, the beat and modulational instabilities are estimated at wavenumbers $kV_A/\Omega_p \approx 0.218$, and $kV_A/\Omega_p \approx 0.075$, respectively.

The simulated magnetic field, density, and bulk velocity fluctuations are firstly averaged in the real space over one of the perpendicular directions (x-direction). After the averaging we obtain a 2D representation of wavevectors with a parallel ($k_\parallel \equiv k_z$) and a perpendicular ($k_\perp \equiv k_y$) component. The power spectrum is constructed in the ($k_\parallel, k_\perp$) domain by Fourier analyzing of the 2D spatial averaged fluctuations. In our setup, the Alfvén pump wave and the field-aligned compressional- and Alfvén-daughter waves have the Fourier modes ($m_\parallel, m_\perp$) with values of (10,0), (18,0), and (-8,0), respectively ($m_{\parallel(\perp)} = k_{\parallel(\perp)}L/2\pi$), accordingly to the wave-wave coupling rules. The negative sign expresses the backward (antiparallel) propagation of the Alfvén daughter wave. In our astrophysical scenario, the Alfvén pump wave is propagating away from the Sun.

## 3   Results

Fig. 2 (left panel) shows the time evolution of the antisunward-propagating Alfvén pump (10,0), the compressional daughter (18,0), and the sunward-propagating Alfvén daughter (-8,0) modes excited by the field-aligned parametric decay. The linear growing of the daughter modes terminates close before time $t\Omega_p$ =300. At later times, the increasing is very slow while the pump wave remains stronger until the end of the simulation time. Besides the field-aligned decay, moderate-oblique propagating daughter waves are excited. Here we show, plotted by dashed and dotted lines, the oblique modes with perpendicular Fourier numbers $m_\perp = 8$, and $m_\perp = 12$. The growth rates are computed as slopes of the density modes, $\gamma = 1/|\delta\rho|d/dt|\delta\rho|$.

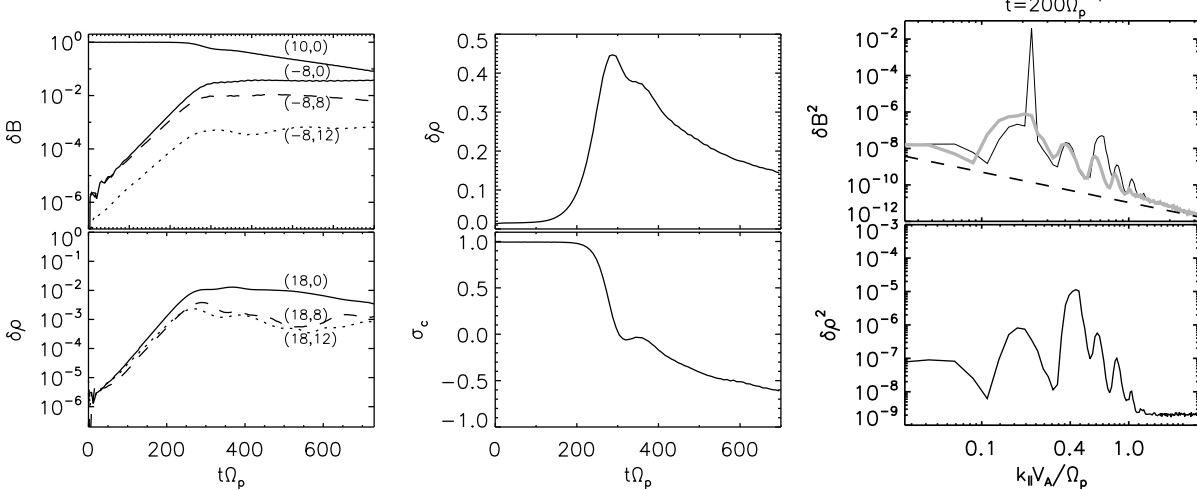

**Figure 2.** Left: Time evolution of the Alfvén pump mode (10,0), the field-aligned Alfvén daughter mode (-8,0), and the ion acoustic daughter mode (18,0) are given by solid lines in the top and bottom panels, respectively. Two moderate obliquely-propagating daughter waves are also drawn by dashed and dotted lines. Middle: Time evolution of the rms density fluctuations (top) and the cross-helicity (bottom). Right: Power spectrum in the $k_\parallel$ wavenumber domain for the decomposed magnetic field (top) and density (bottom) fluctuations before the nonlinear saturation of the decay instability. The sunward (antisunward) propagating left-handed fluctuations are drawn in black (gray). The dashed line describes power laws of the normalized wavenumber $k_\parallel^{-5/3}$.

The field-aligned growth rate $\gamma_\parallel$ has a value of 0.036 close to the above estimated $\gamma_{th}$. **The growth rates for the moderate oblique daughter waves have close values to the parallel daughter mode. Viñas and Goldstein (1991b) show that there is a trend to eliminate/reduce the differences between the growth rates of the oblique and field-aligned decay instabilities while plasma beta values are decreasing to lower values. Following Viñas and Goldstein (1991a, b) analytical treatment,**
5   **our calculations indeed reveal that the oblique growth rates (at propagation angles of 10, and 20 deg) have close values with the field-aligned growth rate for $\beta$=0.01 thus proving the predicted tendency of merging at low beta values.**

The root-mean-squared (rms) density fluctuations $\delta\rho$ and the cross-helicities $\sigma_c$ are represented as a function of time in Fig. 2 (middle panels). The normalized cross-helicity defined as $\sigma_c = (<\delta\boldsymbol{B}\cdot\delta\boldsymbol{u}>)/(<\delta\boldsymbol{B}>^2 + <\delta\boldsymbol{u}>^2)$ where $<\delta\boldsymbol{B}>$ are the averaged magnetic field fluctuations normalized to the mean field $B_0$, while $<\delta\boldsymbol{u}>$ are the averaged bulk velocity fluctuations normalized to the Alfvén velocity $v_A$, has in our study a value of +1 for the antisunward-propagating Alfvén pump wave.

When the decay instability is set on, the density fluctuations start to increase and the cross-helicity starts to decrease. At a time just before t= 300 $\Omega_p^{-1}$, the compressional fluctuations reach a maximum value associated with the saturation of the instability while the sharp decreasing of the cross-helicity terminates within a narrow plateau. A weaker attenuation of the cross-helicity continues down to the value of -0.5 at the latest time of simulation. **In earlier studies concerning the parametric**
15   **decay (e.g., Del Zanna et al., 2001), the cross-helicity of the waves tends to change from positive to negative cross-helicity**

**values for low beta simulations.** As one can see in Fig. 2, at time $t\Omega_p \approx 300$ when $\sigma_c = 0$, the pump wave is still dominant over the parallel propagating daughter mode, thus suggesting that a broadband spectrum of obliquely-propagating waves is developed.

The power spectrum of magnetic field and density fluctuations is given in Fig. 2 (right panel) at time $t\Omega_p$=200 during the linear growth of the daughter modes. The spectrum of the decomposed magnetic field fluctuations shows left-handed antisunward propagating waves (or right-handed sunward propagating waves) drawn by black solid line and left-handed sunward propagating waves (or right-handed antisunward propagating waves) given by gray solid line. The spectrum of the incompressible sense of magnetic energy $(\delta B_x^2 + \delta B_y^2)$ is dominated by the pump wave at $(kV_A/\Omega_p$=0.218$)$. The first broadband gray peak and the third black peak (counting from the left-hand side of the plot) are localized around the lower and upper wavenumbers of the sideband daughter modes ($m_\parallel = 10 \pm 18$ corresponding to $kV_A/\Omega_p$=0.61, and $kV_A/\Omega_p$=0.17, respectively). The compressional daughter mode is the strongest peak in the density spectrum at the wavenumber close to the predicted value from our analytical study ($kV_A/\Omega_p$=0.385). Its second harmonics can be also identified as the second following peak. At smaller wavenumbers close to the pump wavenumber ($kV_A/\Omega_p$=0.218), additional broadband compressional modes are accompanying the ion acoustic daughter wave. Most of the peaks observed in the magnetic field spectrum correspond to the wave-wave couplings of the pump wave with the fundamental and the harmonics of the daughter compressional wave driven by the decay instability. Exceptions are the peaks localized at $kV_A/\Omega_p \approx 0.4$ which can be related to the beat instability and correspond to the coupling of the pump wave with the compressional mode excited at same wavenumber. According to MHD studies (e.g., Bekhor and Drake, 2003), the beat instability driving a fast magnetosonic mode is efficient at larger ion beta plasmas and moderate oblique propagation angles.

While Fig. 2 based on analyzing the wave spectrum brings evidence of the decay instability, Fig. 3 reports the particle heating process. The upper panels of Fig. 3 present the particle distribution functions in the phase space $z - v_\parallel$ at two different stages for the evolution of decay instability. The time evolution of the velocity distribution functions is usually helpful to emphasize the role of the kinetic regime on the saturation of the instability via particle trapping and wave particle interactions. The left panel of Fig. 3 refers to the early stage just after the saturation of the instability at a time of $t\Omega_\mathrm{p} = 300$ while the right panel corresponds to a later time of $t\Omega_\mathrm{p} = 600$. The proton phase space $z - v_\parallel$ shown in the upper panels of Fig. 3 is similar with former studies. Matteini et al. (2010a) explain the spatial modulation and the modulation in enhancement of the parallel electric field (Fig. 5 in their paper) due to the broader spectrum of ion acoustic waves excited by the large amplitude Alfvén mother wave. The spatial modulation in Fig. 3 is weaker according to the smaller amplitude pump wave used in our simulation. During the saturation of the instability we do not observe the presence of phase-space vortices leading to the formation of a proton beam. This is a consequence of the low values for the electron beta ($\beta$=0.01) and ion beta ($\beta$=0.01). At very low electron temperatures, the contribution of the electron pressure term to the electric field ($\nabla P_e = \nabla n k_B T_e$) is small and the particle density fluctuations are less efficient in coupling to the electric field fluctuations. In a first stage, the protons are accelerated by the parallel electric field produced by the density fluctuations. In the later stage ($t\Omega_\mathrm{p} = 600$), the particles are smoothly heated and the resonant protons accelerated by the ion acoustic waves are mixed with the thermal core of the distribution.

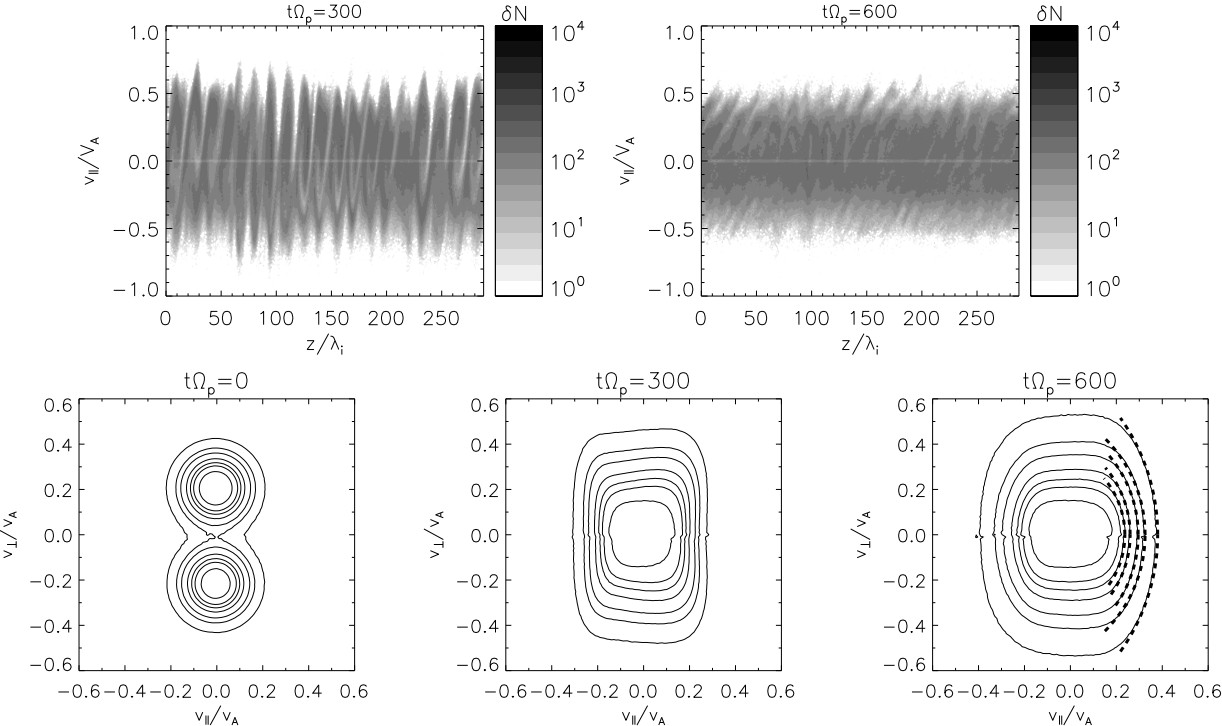

**Figure 3.** Top: Proton phase space $z - v_\parallel$ at time $t\Omega_p = 300$ close to the linear saturation of the decay instability, and at the final time $t\Omega_p = 600$ of the simulation. The particle density is represented in a gray-coded scale with a minimum defined by the lightest nuance of gray. Bottom: Proton reduced distribution functions represented as contour levels (solid line) determined in the plane $(v_\parallel, v_\perp)$ at times $t\Omega_p=0$, $t\Omega_p=300$, and $t\Omega_p=600$. The segments of circle represented by dashed line describe diffusion plateaus (see text).

The lower panels of Fig. 3 report the proton reduced velocity distribution functions constructed in the $(v_\parallel, v_\perp)$ plane obtained at the initial time and at the same simulation epochs as in the upper panels. The reduced velocity distribution functions $f_r(v_\parallel, \pm v_\perp)$ are computed by counting the number of particles $dN = v_\perp dv_\perp dv_\parallel \int f(v_\perp, v_\parallel, \Phi) d\Phi$. Here, $v_\perp = sgn(\pi - \Phi)\sqrt{v_x^2 + v_y^2}$ is the velocity component perpendicular to the mean magnetic field, $v_\parallel \equiv v_z$ is the parallel velocity, and the integral over the azimuthal angle $\Phi = \arctan v_x/v_y$ is done within the interval $[0, 2\pi]$. The contour lines are given for fractions of 60%, 40%, 30%, 20%, 10%, 5%, and 1% of the maximum phase space density from the inner to the outer part of the distribution functions.

Due to the transversal wave field imposed as the initial condition, the velocity distribution functions at time $t\Omega_p=0$ are shifted towards the initial bulk velocities. The deformation of the distribution functions with respect to the Maxwellian shape due to the presence of a wave field of forces can drive an apparent temperature anisotropy, see e.g., Verscharen and Marsch (2011). **The wave effect on the velocity distribution function can be described by a model distribution function which is equivalent to a Maxwellian distribution shifted by the mean fluid-velocity of the particles associated with the local**

magnetic field (e.g., Verscharen and Marsch, 2011; Nariyuki, 2011). In many numerical simulations, e.g., Markovskii et al. (2009), such a modified Maxwellian distribution is used at the initial setup to proper account for the motion of particle in the wave field. The wave field effect on the distribution function by applying the pump wave is however much less important on the distribution functions evaluated at later times because the ratio of the bulk and thermal energies describing the velocity shift of the Maxwellian distribution becomes dominant by considering the wave decay and oscillation of the daughter waves. The symmetrical sets of contour levels with respect to the $v_\perp = 0$ axis are slowly merging with the time evolution of the velocity distributions.

The dashed lines at time $t\Omega_\mathrm{p}$=600 representing segments of circle describe the diffusion plateaus for the resonant protons with positive velocities scattered by backward-propagating daughter waves (see e.g., Isenberg and Lee, 1996),

$$\frac{1}{2}v_\perp^2 + \frac{1}{2}v_\parallel^2 - \int_{v_{0\parallel}}^{v_\parallel} \frac{\omega_{res}(k_\parallel, v_\parallel')}{k_\parallel} dv_\parallel' = \text{constant} \tag{1}$$

where $v_{0\parallel}$ is the initial value of the parallel proton velocity $v_\parallel$ satisfying the following resonance condition,

$$\frac{v_\parallel}{V_A} = \frac{\omega_{res}/\Omega_p}{k_\parallel V_A/\Omega_p} - \frac{1}{k_\parallel V_A/\Omega_p} \tag{2}$$

The above equations are derived within the quasi-linear theory of wave particle interaction in magnetized turbulent plasma (see e.g., Kennel and Engelmann, 1966). The segment of circle represented by the rightmost dashed line in Fig 3 (bottom-right panel) defines the cyclotron diffusion plateau obtained by numerically solving the integral in Eq. (1) including Eq. (2) and the cold plasma dispersion relation modeled by,

$$k_\parallel^2 V_A^2/\Omega_p{}^2 = \frac{\omega(k_\parallel)^2/\Omega_p{}^2}{\sqrt{1 - \omega(k_\parallel)/\Omega_p}} \tag{3}$$

The resulted level plateau has the center localized at a value of $V_{center} \approx -V_A/2$ along the negative axis of the parallel proton velocities. The inner dashed segments are obtained by slightly shifting $V_{center}$ towards $v_\parallel = 0$ axis. Similar diffusion plateaus can be derived for the lefthand side of the plot (with sunward velocity component) corresponding to the resonant scattering of protons by antisunward-propagating waves.

## 4 Discussion: On role of obliquely-propagating waves in proton heating

It is straightforward to notice by comparing the distribution functions at the intermediate ($t\Omega_\mathrm{p}$=300) and final time ($t\Omega_\mathrm{p}$=600), that the contour levels are moderately enlarging both in the parallel and perpendicular directions following the diffusion plateaus of energy conservation driven by the pitch-angle scattering of protons. Observational evidences of diffusion plateaus formed by solar wind protons have been reported starting with the paper of Marsch and Tu (2001) followed by later observational (e.g., Tu and Marsch, 2002; Heuer and Marsch, 2007; Marsch and Bourouaine, 2011; He et al., 2015) or particle-in-cell simulations (e.g., Gary and Saito, 2003) studies. Marsch and Bourouaine (2011) have found that the oblique propagation of waves is the key factor which enables solar wind protons to scatter along the plateau levels of the velocity distribution functions.

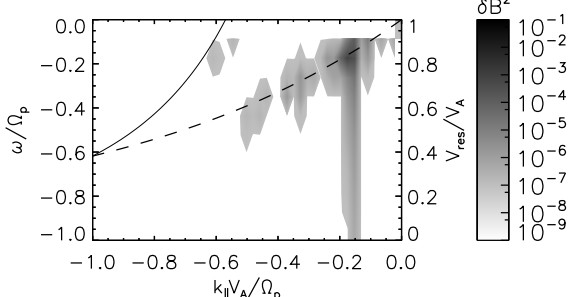

**Figure 4.** Normalized resonance velocity (solid line) of ions in dependence with the wavenumber according to Eq. (2) (see right axis). By dashed line is presented the dispersion relation for cold plasma (see left axis). Overplotted in a gray-coded scale, the frequency - wavenumber spectrum of the magnetic field fluctuations is given at time $t\Omega_p = 600$.

Here we will check whether field-aligned propagating waves can resonate and scatter protons by pitch-angle diffusion mechanism or obliquely propagating waves are necessarily to explain the plateau levels observed in Fig. 3. The resonance velocity defined in Eq. (2) is plotted by solid line along the wavenumber axis in Fig. 4. The right vertical axis shows the values of the resonance velocity in terms of Alfvén velocity $V_A$. The frequency-wavenumber ($\omega - k$) spectrum corresponding
to the sunward-propagating Alfvén daughter waves is determined at time $t\Omega_p = 600$ and represented in a gray-coded scale in same figure. By dashed line is represented the cold plasma dispersion relation. Besides the Alfvén daughter wave observed at wavenumber $kV_A/\Omega_p \approx 0.18$ as the strongest mode driven by the decay instability, a trend of additional normal modes are excited and forming a relatively broadband (turbulent) spectrum at larger wavenumbers and frequencies. The maximum value of the parallel velocity $v_{\parallel max} \approx 0.4V_A$ reported in the velocity distribution functions for the outmost level contour corresponds
to a resonance velocity at the parallel wavenumber $k_{\parallel}V_A/\Omega_p \approx -1$. Due to the lack of waves in this wavenumber region, the observed parallel-propagating daughter waves cannot explain the formation of plateau levels in the velocity distribution functions. The remaining candidates capable to scatter the particles along the segments of circles at moderate parallel velocities are the obliquely-propagating modes. At oblique inclination angles, the dispersion relation branch becomes less tilted with respect to the wavenumber axis while the resonance velocity is shifted towards smaller (absolute) wavenumbers; therefore
the resonance condition is expected to be fulfilled. However, an extensive analysis of the pitch-angle scattering of protons by obliquely-propagating waves in the framework of quasi-linear theory is beyond the objectives of the actual study while the determination of the frequency-wavenumber spectrum of magnetic field fluctuations at oblique angles is hard to be achieved due to the requirement of sufficient spatial resolution.

The important role of the obliquely-propagating daughter waves in perpendicularly heating the particles can be alternatively
emphasized by the comparison with the results obtained from additional simulations carried out by decreasing the dimensionality from the 3-D down to the 2-D and 1-D configurations **while all the other physical and numerical parameters are basically maintained the same.** In the one-dimensional box spatial variations are allowed only in directions parallel and an-

tiparallel to the background magnetic field, whereas in the two-dimensional simulation spatial variations are allowed in both parallel/antiparallel direction and one perpendicular direction. Fig. 5 shows the time evolution of the parallel and perpendicular proton temperatures obtained in the actual and the additional 2-D and 1-D setups. **The temperatures are determined by computing the thermal velocities (as the second-order velocity moment) obtained by subtracting the bulk velocity from the full particle velocities according to the definition of kinetic temperature.** The time evolution of the parallel temperature of protons is similar for all the simulation runs. In contrast, the perpendicular temperature for the 3-D setup starts to increase and becomes larger by a factor of four with respect to the temperatures obtained in the downgraded configurations. The three simulations carried out with same physics in one dimension, two dimensions, and three dimensions demonstrate that the 3-D simulation yields the strongest proton heating.

**With respect to this result we have done a quantification of the differences between the results of the 2-D and 3-D setups in what means the amplitude and slope of the oblique modes. At the linear stage of the instability growing, the Alfvén oblique mode (-8,8) shown in Fig. 2 has close amplitude and slope as the field-aligned daughter mode (-8,0). The correspondent mode in the 2-D setup is about 3 times weaker in amplitude than the field-aligned daughter mode computed in this configuration. The perpendicular projection of the obliquely-propagating density mode (18,8) is more unstable and grows faster than its counterpart from the 2-D setup. On the other hand, the 3-D case opens new channels of the parametric instabilities in the sense that there is a larger population of obliquely propagating waves due to a degree of freedom in the azimuthal direction around the mean magnetic field. As the oblique propagating waves play an important role in the heating process of low beta plasmas, the enhanced oblique daughter waves lead to a more efficient ion heating in the perpendicular direction.**

**Viñas and Goldstein (1991a, b) discovered new channels of parametric instabilities when the dimension of the analyzed MHD system is increased from 1-D to 2-D thus allowing that the daughter and side-band modes to obliquely propagate with respect to the field-aligned Alfvén pump wave. Some of these new instabilities, e.g., the filamentation instability, have been identified in later MHD simulations (Ghosh et al. 1993, 1994) by comparison with the analytical predictions of Viñas and Goldstein (1991b). To emphasize new instabilities particularly enhanced in 3-D with respect to 2-D setups an analytical study is needed by considering both the oblique and azimuthal propagation angles of daughter modes. However, analytic treatments concerning oblique instabilities developed by parallel propagating Alfvén pump waves have no longer continued since Viñas and Goldstein (1991a, b) papers. An outcome of these former theoretical studies confirmed in the present numerical study is that the obliquely-propagating daughter waves are stronger and play a more important role in the dynamics of the heating process in low beta plasmas than the 1-D field-aligned pump case.**

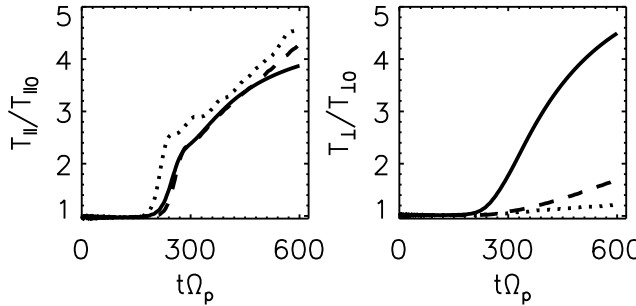

**Figure 5.** Time evolution of the parallel ($T_{\parallel}$) and perpendicular ($T_{\perp}$) proton temperatures. The dashed and dotted lines represent the result obtained from downgraded 2-D and 1-D configurations. The temperatures are normalized to their initial values.

## 5    Conclusions

We performed for the first time a 3-D hybrid simulation study on the plasma heating problem associated with the parametric decay. The analysis of wave-wave coupling driven by decay instability and the time evolution of the main wave modes and cross-helicity, bring evidence that obliquely-propagating waves are early excited by the field-aligned Alfvén pump wave.

5    We draw the following conclusions:

1. The pitch-angle scattering is efficient in 3 dimensions and the plasma becomes heated more quickly by the obliquely-propagating daughter waves. The stochastic heating can be verified by the upcoming solar wind measurements by Parker Solar Probe and Solar Orbiter. A temperature rise by a factor of about 4 is obtained in our 3-D hybrid simulation study by the time of 300 to 600 ion gyroperiods. When applying the mapping of the elapsed time into the radial distance from the Sun advected by the solar wind (Comişel et al., 2015), we obtain a radial distance of about 0.1 to 0.2 AU from the Sun at the time of 300 to 600 gyroperiods.

2. Thermal core particle population is effectively heated in 3 dimensions as well in the longitudinal (to the wavevector) and parallel (to the mean magnetic field) direction by the field-aligned and the obliquely-propagating sound waves out of the parametric decay, comforting the lessons from the earlier studies (Laveder et al., 2002; Gao et al., 2013).

15    Needless to say, our conclusions are limited to a beta parameter of 0.01. 3-D hybrid simulations provide more realistic predictions for the wave-heating problem in nonlinear space plasma dynamics. We propose to study the following items to extend the 3-D simulations. Viñas and Goldstein (1991a, b), Ghosh et al. (1993, 1994) and Ghosh and Goldstein (1994) discovered the importance of the oblique waves in the parametric decay in dependence of the beta parameter. **One may expect that plasma beta parameter $\beta$ could be the key factor in controlling the perpendicular ion heating driven by the**
20    **parametric decay instability in space plasmas.**

*Competing interests.* The authors declare that they have no conflict of interest.

*Acknowledgements.* This work is financially supported by a grant of the Deutsche Forschungsgemainschaft (DFG grant MO539/20-1). HC acknowledges the hospitality at University of Toyama for hosting the research visit. We acknowledge John von Neumann Institute for Computing (NIC) by providing computing time on the supercomputer JURECA at Juelich Supercomputer Centre (JSC).

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
