# Peer review of "On heating of solar wind protons by the parametric decay of large amplitude Alfvén waves"

_Annales Geophysicae, 2018_

## Referee Comment (RC1) · Anonymous Referee #1 · 13 Feb 2018

This manuscript describes a hybrid kinetic simulation study of a parametric instability in which two counter-propagating Alfven waves couple with a spectrum of ion acoustic modes to transfer fluctuation energy from the former into the latter. This is an interesting configuration to examine, but the manuscript is incomplete because it does not clearly discuss the physical consequences of the computation.

The central problem here is that Figure 3 and the associated discussion is not clearly defined. I do not agree that Figure 3 shows that the velocity distributions are "different for the three analyzed systems";

to my eye the six panels of Figure 3 are qualitatively all the same.

I disagree that "the final distribution functions for the 3D system. . .report a larger perpendicular acceleration."

To substantiate this claim, the authors should do the velocity integrations to compute $T_\parallel$ and $T_\perp$ as functions of time through the simulation. The sentence "...these arcs coincide with the most obliquely parts of contour lines while the outer contours are better overlapped than the inner ones" is confusing, and I find the subsequent discussion through page 6 difficult to follow. Add the t=0 contours to Figure 3, and compute the $T_\parallel$ and $T_\perp$ values as functions of time to quantify the statements in the discussion.

A central point of this manuscript is that the 3D simulations yield better results than the corresponding 1D and 2D results. This point should be made in the Abstract and repeated in the Conclusion section.

The proton velocity distributions measured from spacecraft in the fast solar wind often show a beam component and a core component with different relative densities and relative flow velocities parallel to the background magnetic field. Figure 3 of this manuscript shows two proton components of equal densities with relative flow velocities perpendicular to $B\_0$. The Abstract claims the results of the simulations are in agreement with in situ measurements; to justify this claim, the authors need to explain these differences.

Title: There is no discussion or demonstration of wave "breaking" here, this word should be deleted from the title.

Page 2, Line 9: Delete "so".

Page 2, Line 23: "Low-beta"?

Page 2, Line 31: "in directions perpendicular"...

Page 3, Line 12: Replace "circularly" with "circular".

Page 3, Line 15: Replace "transversal" with "fluctuating".

Page 3, Lines 18-19: "The parametric decay modeled here is a 3-wave process involving a large-amplitude monochromatic Alfven pump wave propagating parallel to B_0, a spectrum of electrostatic ion acoustic waves also at parallel propagation, and a spectrum of Alfven daughter waves at anti-parallel propagation."

Page 3, Line 22: Delete "linear" (saturation is a nonlinear process).

Page 3, Line 23: Delete "nonlinear"; it is unnecessary.

Page 3, Line 33: "…and the lower panels correspond to the end of the simulation (t Omega_cp =600)."

Page 6: Insert the definitions of the solid lines and the dashed lines in the caption to Figure 3.

---

## Referee Comment (RC2) · Anonymous Referee #2 · 26 Feb 2018

The manuscript #angeo-2018-14 by Comişel et al. discusses heating of protons during the course of nonlinear evolution of a large amplitude Alfven wave in the solar wind. The author used a hybrid code to conduct 1D, 2D, and 3D simulations all started with a circularly polarized pump Alfven wave imposed at the initial condition. The simulation results showed that an efficient proton heating in the perpendicular direction occurs in 3D, but not in 1D and 2D. I think the finding itself is interesting and the paper is potentially worth for publication. However, the authors' discussion does not explain how the discrepancy arises between the different simulations. Also, there is a plenty of room for improvement in the quality of presentation. Therefore, I believe that substantial revisions need to be made on the manuscript before the publication.

Major Issues

[Figure]

============

The simulation results suggested that pitch-angle scatterings by both the pump and daughter Alfven waves are more important than the heating in the parallel direction via damping of ion sound waves in 3D. In contrast, 1D and 2D results suggested that the pitch-angle scatterings are less important compared to 3D. It is not entirely clear how the difference arises. If you look at Figure 1, you can clearly see that the ion sound wave amplitudes do not change much between 2D and 3D. On the other hand, the amplitude of the daughter Alfven wave in 3D is much less than that in 2D. Apparently, this contradicts with the behavior seen in the distribution function.

The 3D simulation presented in the paper is quite large and should contain a lot of information. My impression is that the authors did not make use of the benefits of the large-scale simulation. I would encourage the authors to conduct more detailed analysis and draw more physically sound and grounded conclusions based on the data.

Minor Issues

============

* The authors used beta = 0.01, which is very small in comparison with the solar wind at 1 AU. Please consider to state the motivation for adopting this particular value. Also, the author may think it better to include discussion on the dependence of the plasma beta.

* P.2, L. 32: "the conditions of beta plasmas" does not make sense.

* P.3, L. 29: The sentence ending with "due to most probably the small value of electron beta used in simulation" needs more clarification.

* P.4, L. 25: What the authors meant by the sentence "We see that these arcs..." is not clear. I can guess what you wanted to mention, but in general, it is not a good idea to let the readers guess the meanings.

* Figure 2: Are the color scales linear or logarithmic? Are they the same or different between the four panels? It is difficult for me to see behaviors in the low-energy part of the distribution function. Is this the authors' intention?

* Figure 3: It would be better to add a description for the dashed lines in the caption. The authors' definition of v_perp should never become negative. You need a more clear description what the distribution functions in Figure 3 represent.

---

## Author Comment (AC1) · 16 Apr 2018

**Reply to referee comments**

On heating of solar wind protons by the breaking of large amplitude Alfvén waves

Manuscript ID: angeo-2018-14 (AngeoComm)
H. Comişel, Y. Nariyuki, Y. Narita, and U. Motschmann

Thank you very much for reading the manuscript and raising helpful comments and suggestions.

[Figure]

- *This manuscript describes a hybrid kinetic simulation study of a parametric instability in which two counter-propagating Alfven waves couple with a spectrum of ion acoustic modes to transfer fluctuation energy from the former into the latter. This is an interesting configuration to examine, but the manuscript is incomplete because it does not clearly discuss the physical consequences of the computation.*

  *The central problem here is that Figure 3 and the associated discussion is not clearly defined. I do not agree that Figure 3 shows that the velocity distributions are "different for the three analyzed systems"; to my eye the six panels of Figure 3 are qualitatively all the same."*

Reply:

Figure 3 has been updated by including a missing term ($1/v_\perp$) used to properly compute the velocity distribution function in cylindrical coordinates. The velocity distribution function $f$ is computed by counting the number of particles $dN = f(v_\perp, v_\parallel, \Phi)dV$ in the volume element $dV = v_\perp dv_\perp dv_\parallel d\Phi$, and by integrating over the azimuthal $\Phi$ angle. Here, $v_\perp = \sqrt{v_{\perp 1}^2 + v_{\perp 2}^2}$ is the velocity component perpendicular to the mean magnetic field, $v_\parallel$ is the parallel velocity, and the angle $\Phi = \arctan v_{\perp 1}/v_{\perp 2}$ gives the sign of $v_\perp$. At times ($t\Omega_p$=300; $t\Omega_p$=600), the updated plots are slightly different from those shown in former Fig. 3.

We accept the referee criticism that there are no obvious differences between the velocity distributions shown in Fig. 3 (or in new Fig.3). However, one can notice a distinct trend of evolution for the distributions between the intermediate ($t\Omega_p$=300) and the final time of simulation ($t\Omega_p$=600). At the final time, the contour levels in the 3D simulation are moderately enlarging both in the parallel and perpendicular directions by following the contour levels of energy conservation

driven by the pitch-angle scattering of protons. The contour levels in 1D and 2D runs, in contrast, are developing mainly towards parallel direction while their initial perpendicular displacement is removing with time elapsing.

Changes in the manuscript:

– Page 5: Line 6 to Line 10:
"The velocity distribution function $f$ is computed by counting the number of particles $dN = f(v_\perp, v_\parallel, \Phi)dV$ in the volume element $dV = v_\perp dv_\perp dv_\parallel d\Phi$, and by integrating over the azimuthal $\Phi$ angle. Here, $v_\perp = \sqrt{v_{\perp 1}^2 + v_{\perp 2}^2}$ is the velocity component perpendicular to the mean magnetic field, $v_\parallel$ is the parallel velocity, and the angle $\Phi = \arctan v_{\perp 1}/v_{\perp 2}$ gives the sign of $v_\perp$."

– Page 5: Line 11 to Line 14:
"Due to the transversal wave field imposed at the initial condition, the velocity distribution functions are rigid shifted towards the initial bulk velocity $\pm u_\perp$, see e.g., Verscharen2011, Nariyuki2011. The two symmetrical sets of contour levels with respect to the $v_\perp = 0$ axis are slowly merging with the time evolution and at time t$\Omega_p$=300 there are no remnants of the rigid displacement observed at the initial time."

– Former manuscript, Page 5: Line 1 to Line 3 - deleted.

– Page 6: Line 6 to Line 11:
"Although there are no obvious differences between the velocity distributions in the three different setups, one can notice a distinct trend of evolution for the distributions between the intermediate ($t\Omega_p$=300) and the final time of simulation ($t\Omega_p$=600). At the final time, the contour levels in the 3D simulation are moderately enlarging both in the parallel and perpendicular directions by following the contour levels of energy conservation driven by the pitch-angle scattering of protons. The contour levels in 1D and 2D runs, in contrast, are developing mainly towards parallel direction while their initial perpendicular displacement is removing with time elapsing."

- *I disagree that "the final distribution functions for the 3D system ... report a larger perpendicular acceleration. To substantiate this claim, the authors should do the velocity integrations to compute $T_{||}$ and $T_\perp$ as functions of time through the simulation.*

Reply:
The time evolutions of $T_{||}$ and $T_\perp$ determined during the simulation runs are shown in new Figure 4. Figure 4 reports that the perpendicular temperature (normalized to its initial value) achieved in the 3D system at the final time ($t\Omega_p$=600) is about two times larger than that one from the 2D system.

Changes in the manuscript:

- Page 6: Line 11 to Line 14:
  "The more efficient heating of plasma in the 3D system is consistent with the time evolution of the ion temperature shown in Fig. 4. The particles experience a similar parallel heating while the perpendicular temperature achieved in the 3D system (solid line) dominates by a factor of two or more the corresponding values obtained in the 1D (dotted line) and 2D (dashed line) simulations. "

- *The sentence "... these arcs coincide with the most obliquely parts of contour lines while the outer contours are better overlapped than the inner ones" is confusing, and I find the subsequent discussion through page 6 difficult to follow.*

Reply:
This sentence and the following ones (Page 5, Line 16 to Page 6, Line 9 in the former manuscript) have been deleted. Instead, we introduced a new comment discussed above (Page 6, Line 6 to Line 11). The next paragraph in the former manuscript (Page 6, Line 10 to Line 14) was replaced by the following comment.

Changes in the manuscript:

- Page 6: Line 15 to Line 29:
  "These results suggest that the damping of the ion sound waves excited by the field aligned parametric decay is the main mechanism of plasma heating in the 1D and 2D systems. In the 3D system, the protons are also heated in the perpendicular direction by the cyclotron damping of waves. The ions are perpendicular scattered by the field aligned and the oblique developed Alfvén daughter waves.

  Figure 1 shows that the amplitude of the anti-parallel propagating Alfvén daughter wave decreases with increasing spatial dimension, while the level of the density fluctuation is similar. The power spectrum $\delta \vec{B}^2(k_\parallel)$ is obtained by the Fourier transformation of the averaged magnetic field, $\delta \tilde{\vec{B}}(r_\parallel) = \int \delta \vec{B}(r_{\perp 1}, r_{\perp 2}, r_\parallel) dr_{\perp 1} dr_{\perp 2}$, in the assumption of strictly parallel wave propagation. Any deviation from the parallel direction will conduct to a reduction of the $E^D$ amplitude of the daughter mode. A slight obliquity (several degrees) of the daughter wave mode is noticed in the 3D system but the rea-
son remains unclear within the simulation work here and needs further investigations. The pitch-angle scattering and the perpendicular temperature increase observed in the time evolutions of the velocity distribution functions and temperatures, respectively, suggest that the Alfvén daughter waves are in cyclotron resonance with protons and the wave-particle interaction could explain the deviation in the propagation angle and the stronger damping of the daughter waves in the 3D system. A detailed spectral analyzing of the oblique wave modes developed in the decay process based on the 2D reduced magnetic field spectrum will be subject for a further study."

- *Add the t=0 contours to Figure 3, and compute the $T_{||}$ and $T_{\perp}$ values as functions of time to quantify the statements in the discussion.*

Reply:
Done.

Changes in the manuscript:

    – Fig. 3 is updated with time t=0.

    – New Fig. 4 shows the time evolution of temperatures.

- *A central point of this manuscript is that the 3D simulations yield better results than the corresponding 1D and 2D results. This point should be made in the Abstract and repeated in the Conclusion section.*

Reply:
We thank the referee for this suggestion.

Changes in the manuscript:

– Abstract, Page 1: Line 3 to Line 6:
  "The comparison made among different spatial dimensions proves that the three-dimensional simulation exhibits more efficient heating. Plasma is heated parallel to the mean magnetic field by the damping of the ion acoustic waves while being heated perpendicular by the cyclotron resonance and damping of protons by Alfvén daughter waves."

– Conclusion, Page 6: Line 33 to Page 7 Line 4:
  "By comparing the wave modes and proton velocity distribution functions in 1D, 2D, and 3D systems, we conclude that the plasma is heated more efficient in the 3D system, thus proving that the 3D simulations yield better results than the corresponding 1D and 2D results. Parallel heating of plasma is provided by the damping of ion sound waves while perpendicular heating is given by the perpendicular scattering of protons by the field aligned and the oblique developed Alfvén daughter waves."

• *The proton velocity distributions measured from spacecraft in the fast solar wind often show a beam component and a core component with different relative densities and relative flow velocities parallel to the background magnetic field. Figure 3 of this manuscript shows two proton components of equal densities with relative flow velocities perpendicular to $B_0$. The Abstract claims the results of the simulations are in agreement with in situ measurements; to justify this claim, the authors need to explain these differences.*
Reply:
We reformulated the sentence as follows: "In the solar wind context, the antisunward part of the core component of the proton velocity distributions is controlled by the sunward-propagating waves driven by the parametric decay."

Changes in the manuscript:

– Page 1: Line 6 to Line 7
"In the solar wind context, the antisunward part of the core component of the proton velocity distributions is controlled by the sunward-propagating waves driven by the parametric decay."

- *Title: There is no discussion or demonstration of wave "breaking" here, this word should be deleted from the title.*

Changes in the manuscript:

– Title: We replaced "breaking" by "parametric decay".

- *Page 2, Line 9: Delete "so".*

Reply:
Done.

Changes in the manuscript:
– Page 2: Line 9.

• *Page 2, Line 23: "Low-beta"?*

Reply:
Done.

Changes in the manuscript:

– Page: 2, Line 23: "low-beta plasmas."

• *Page 2, Line 31: "in directions perpendicular" ...*

Reply:
Done.

Changes in the manuscript:

– Page: 2, Line: 31.
   "... in directions perpendicular to the mean magnetic field."

• *Page 3, Line 12: Replace "circularly" with "circular".*

Reply:
Done.

Changes in the manuscript:

    – Page 3, Line 16.

• *Page 3, Line 15: Replace "transversal" with "fluctuating".*

Reply:
Done.

Changes in the manuscript:

    – Page 3: Line 19
      "The initial fluctuating magnetic field ..."

• *Page 3, Lines 18-19: "The parametric decay modeled here is a 3-wave process involving a large-amplitude monochromatic Alfven pump wave propagating parallel to $B_0$, a spectrum of electrostatic ion acoustic waves also at parallel propagation, and a spectrum of Alfven daughter waves at anti-parallel propagation."*

Reply:
Done.

Changes in the manuscript:

– Page 3: Line 23 - 25
"The parametric decay modeled here is a three-wave process involving a large-amplitude monochromatic Alfvén pump wave propagating parallel to $B_0$, a spectrum of electrostatic ion acoustic waves also at parallel propagation, and a spectrum of Alfven daughter waves at anti-parallel propagation."

• *Page 3, Line 22: Delete "linear" (saturation is a nonlinear process).*

Reply:
Done.

Changes in the manuscript:

– Page 3: Line 28.

• *Page 3, Line 23: Delete "nonlinear"; it is unnecessary.*

Reply:
Done.

Changes in the manuscript:

[Figure]

– Page 3: Line 29.

• *Page 3, Line 33: "... and the lower panels correspond to the end of the simulation (t $\Omega_{cp}$ =600)."*

Reply:
Done.

Changes in the manuscript:

– Page 4: Line 7
  "... and the lower panels correspond to the end of the simulation (t $\Omega_p$ =600)."

• *Page 6: Insert the definitions of the solid lines and the dashed lines in the caption to Figure 3.*

Reply:
Done.

Changes in the manuscript:

– Caption of Fig. 3:
  "... The dashed lines describe the locus $(v_{\parallel}, v_{\perp})$ of the particle velocities

$v = \sqrt{(v_\parallel - V_{ph})^2 + v_\perp^2}$ where their energy is conserved in the wave frame. Here $V_{ph}$ is the phase speed of the Alfvén wave."

Other changes in the manuscript:

– Page: 3, Line 11 to line 15
"The value of beta parameter 0.01 is set in order to keep the same value of beta in all the 1-D, 2-D, and 3-D setups for the purpose of comparison, and moreover, a lower value of beta (such as 0.01) is not irrelevant from the solar wind studies. In fact, the solar wind plasma originates in the corona and the low-beta plasmas are more representative in the inner heliosphere. Therefore, we regard our numerical studies not only for understanding the solar wind but also for understanding the solar corona."

– Page 4: Line 4 to Line 6
"The time evolution of the velocity distribution functions is usually helpful to emphasize the role of the kinetic regime on the saturation of the instability via particle trapping and wave particle interactions."

– Page 4: Line 13 to Line 16
"This is a consequence of the low values for the electron beta ($\beta$=0.01) and ion beta ($\beta$=0.01) used in the simulation. At very low electron temperatures, the contribution of the electron pressure term to the electric field ($\nabla P_e = \nabla n k_B T_e$) is small and the particle density fluctuations $n$ are less efficient in coupling to the electric field fluctuations."

**Supplement:**

**Reply to referee comments**

On heating of solar wind protons by the breaking of large amplitude Alfvén waves

Manuscript ID: angeo-2018-14 (AngeoComm)
H. Comişel, Y. Nariyuki, Y. Narita, and U. Motschmann

5   Thank you very much for reading the manuscript and raising helpful comments and suggestions.

– *This manuscript describes a hybrid kinetic simulation study of a parametric instability in which two counter-propagating Alfven waves couple with a spectrum of ion acoustic modes to transfer fluctuation energy from the former into the latter. This is an interesting configuration to examine, but the manuscript is incomplete because it does not clearly discuss the physical*
10   *consequences of the computation.*

*The central problem here is that Figure 3 and the associated discussion is not clearly defined. I do not agree that Figure 3 shows that the velocity distributions are "different for the three analyzed systems"; to my eye the six panels of Figure 3 are qualitatively all the same.*

Reply:

Figure 3 has been updated by including a missing term ($1/v_\perp$) used to properly compute the velocity distribution function in cylindrical coordinates. The velocity distribution function $f$ is computed by counting the number of particles $dN = f(v_\perp, v_\parallel, \Phi)dV$ in the volume element $dV = v_\perp dv_\perp dv_\parallel d\Phi$, and by integrating over the azimuthal $\Phi$ angle. Here, $v_\perp = \sqrt{v_{\perp 1}^2 + v_{\perp 2}^2}$
20   is the velocity component perpendicular to the mean magnetic field, $v_\parallel$ is the parallel velocity, and the angle $\Phi = \arctan v_{\perp 1}/v_{\perp 2}$ gives the sign of $v_\perp$. At times ($t\Omega_p$=300; $t\Omega_p$=600), the updated plots are slightly different from those shown in former Fig. 3.

We accept the referee criticism that there are no obvious differences between the velocity distributions shown in Fig. 3 (or in new Fig.3). However, one can notice a distinct trend
25   of evolution for the distributions between the intermediate ($t\Omega_p$=300) and the final time of simulation ($t\Omega_p$=600). At the final time, the contour levels in the 3D simulation are moderately enlarging both in the parallel and perpendicular directions by following the contour levels of energy conservation driven by the pitch-angle scattering of protons. The contour levels in 1D and 2D runs, in contrast, are developing mainly towards parallel direction while their initial
30   perpendicular displacement is removing with time elapsing.

Changes in the manuscript:

– Page 5: Line 6 to Line 10:
"The velocity distribution function $f$ is computed by counting the number of parti-
35   cles $dN = f(v_\perp, v_\parallel, \Phi)dV$ in the volume element $dV = v_\perp dv_\perp dv_\parallel d\Phi$, and by inte-
grating over the azimuthal $\Phi$ angle. Here, $v_\perp = \sqrt{v_{\perp 1}^2 + v_{\perp 2}^2}$ is the velocity compo-
nent perpendicular to the mean magnetic field, $v_\parallel$ is the parallel velocity, and the angle
$\Phi = \arctan v_{\perp 1}/v_{\perp 2}$ gives the sign of $v_\perp$."
– Page 5: Line 11 to Line 14:
40   "Due to the transversal wave field imposed at the initial condition, the velocity distri-
bution functions are rigid shifted towards the initial bulk velocity $\pm u_\perp$, see e.g., Ver-
scharen2011, Nariyuki2011. The two symmetrical sets of contour levels with respect to

the $v_\perp = 0$ axis are slowly merging with the time evolution and at time $t\Omega_p$=300 there
are no remnants of the rigid displacement observed at the initial time."

45     – Former manuscript, Page 5: Line 1 to Line 3 - deleted.

    – Page 6: Line 6 to Line 11:
"Although there are no obvious differences between the velocity distributions in the three
different setups, one can notice a distinct trend of evolution for the distributions between
50     the intermediate ($t\Omega_p$=300) and the final time of simulation ($t\Omega_p$=600). At the final time,
the contour levels in the 3D simulation are moderately enlarging both in the parallel and
perpendicular directions by following the contour levels of energy conservation driven
by the pitch-angle scattering of protons. The contour levels in 1D and 2D runs, in con-
trast, are developing mainly towards parallel direction while their initial perpendicular
55     displacement is removing with time elapsing."

– *I disagree that the final distribution functions for the 3D system ... report a larger perpendic-
ular acceleration. To substantiate this claim, the authors should do the velocity integrations
to compute $T_\parallel$ and $T_\perp$ as functions of time through the simulation.*

Reply:
60 The time evolutions of $T_\parallel$ and $T_\perp$ determined during the simulation runs are shown in new
Figure 4. Figure 4 reports that the perpendicular temperature (normalized to its initial value)
achieved in the 3D system at the final time ($t\Omega_p$=600) is about two times larger than that one
from the 2D system.

Changes in the manuscript:
65

    – Page 6: Line 11 to Line 14:
"The more efficient heating of plasma in the 3D system is consistent with the time evo-
lution of the ion temperature shown in Fig. 4. The particles experience a similar parallel
heating while the perpendicular temperature achieved in the 3D system (solid line) dom-
70     inates by a factor of two or more the corresponding values obtained in the 1D (dotted
line) and 2D (dashed line) simulations. "

– *The sentence "... these arcs coincide with the most obliquely parts of contour lines while
the outer contours are better overlapped than the inner ones" is confusing, and I find the
subsequent discussion through page 6 difficult to follow.*

75 Reply:
This sentence and the following ones (Page 5, Line 16 to Page 6, Line 9 in the former
manuscript) have been deleted. Instead, we introduced a new comment discussed above (Page
6, Line 6 to Line 11). The next paragraph in the former manuscript (Page 6, Line 10 to Line
14) was replaced by the following comment.

80 Changes in the manuscript:

– Page 6: Line 15 to Line 29:

"These results suggest that the damping of the ion sound waves excited by the field aligned parametric decay is the main mechanism of plasma heating in the 1D and 2D systems. In the 3D system, the protons are also heated in the perpendicular direction by the cyclotron damping of waves. The ions are perpendicular scattered by the field aligned and the oblique developed Alfvén daughter waves.

Figure 1 shows that the amplitude of the anti-parallel propagating Alfvén daughter wave decreases with increasing spatial dimension, while the level of the density fluctuation is similar. The power spectrum $\delta B^2(k_\parallel)$ is obtained by the Fourier transformation of the averaged magnetic field, $\delta \tilde{B}(r_\parallel) = \int \delta B(r_{\perp 1}, r_{\perp 2}, r_\parallel) dr_{\perp 1} dr_{\perp 2}$, in the assumption of strictly parallel wave propagation. Any deviation from the parallel direction will conduct to a reduction of the $E^D$ amplitude of the daughter mode. A slight obliquity (several degrees) of the daughter wave mode is noticed in the 3D system but the reason remains unclear within the simulation work here and needs further investigations. The pitch-angle scattering and the perpendicular temperature increase observed in the time evolutions of the velocity distribution functions and temperatures, respectively, suggest that the Alfvén daughter waves are in cyclotron resonance with protons and the wave-particle interaction could explain the deviation in the propagation angle and the stronger damping of the daughter waves in the 3D system. A detailed spectral analyzing of the oblique wave modes developed in the decay process based on the 2D reduced magnetic field spectrum will be subject for a further study."

– *Add the t=0 contours to Figure 3, and compute the $T_{\parallel}$ and $T_{\perp}$ values as functions of time to quantify the statements in the discussion.*

Reply:
Done.

Changes in the manuscript:

– Fig. 3 is updated with time t=0.
– New Fig. 4 shows the time evolution of temperatures.

– *A central point of this manuscript is that the 3D simulations yield better results than the corresponding 1D and 2D results. This point should be made in the Abstract and repeated in the Conclusion section.*

Reply:
We thank the referee for this suggestion.

Changes in the manuscript:

– Abstract, Page 1: Line 3 to Line 6:
"The comparison made among different spatial dimensions proves that the three-dimensional

simulation exhibits more efficient heating. Plasma is heated parallel to the mean magnetic field by the damping of the ion acoustic waves while being heated perpendicular by the cyclotron resonance and damping of protons by Alfvén daughter waves."

– Conclusion, Page 6: Line 33 to Page 7 Line 4:
"By comparing the wave modes and proton velocity distribution functions in 1D, 2D, and 3D systems, we conclude that the plasma is heated more efficient in the 3D system, thus proving that the 3D simulations yield better results than the corresponding 1D and 2D results. Parallel heating of plasma is provided by the damping of ion sound waves while perpendicular heating is given by the perpendicular scattering of protons by the field aligned and the oblique developed Alfvén daughter waves."

– *The proton velocity distributions measured from spacecraft in the fast solar wind often show a beam component and a core component with different relative densities and relative flow velocities parallel to the background magnetic field. Figure 3 of this manuscript shows two proton components of equal densities with relative flow velocities perpendicular to $B_0$. The Abstract claims the results of the simulations are in agreement with in situ measurements; to justify this claim, the authors need to explain these differences.*

Reply:
We reformulated the sentence as follows: "In the solar wind context, the antisunward part of the core component of the proton velocity distributions is controlled by the sunward-propagating waves driven by the parametric decay."

Changes in the manuscript:

– Page 1: Line 6 to Line 7
"In the solar wind context, the antisunward part of the core component of the proton velocity distributions is controlled by the sunward-propagating waves driven by the parametric decay."

– *Title: There is no discussion or demonstration of wave breaking here, this word should be deleted from the title.*

Changes in the manuscript:

– Title: We replaced "breaking" by "parametric decay".

– *Page 2, Line 9: Delete so.*

Reply:
Done.

Changes in the manuscript:

– Page 2: Line 9.

– *Page 2, Line 23: Low-beta?*

Reply:

160     Done.

Changes in the manuscript:

    – Page: 2, Line 23: "low-beta plasmas."

– *Page 2, Line 31: in directions perpendicular ...*

165     Reply:
    Done.

Changes in the manuscript:

    – Page: 2, Line: 31.
170       "... in directions perpendicular to the mean magnetic field."

– *Page 3, Line 12: Replace circularly with circular.*

    Reply:
    Done.

Changes in the manuscript:

175

    – Page 3, Line 16.

– *Page 3, Line 15: Replace transversal with fluctuating.*

    Reply:
    Done.

180     Changes in the manuscript:

    – Page 3: Line 19
      "The initial fluctuating magnetic field ..."

185    – *Page 3, Lines 18-19: "The parametric decay modeled here is a 3-wave process involving a large-amplitude monochromatic Alfven pump wave propagating parallel to $B_0$, a spectrum of electrostatic ion acoustic waves also at parallel propagation, and a spectrum of Alfven daughter waves at anti-parallel propagation."*

Reply:
Done.

190    Changes in the manuscript:

– Page 3: Line 23 - 25
"The parametric decay modeled here is a three-wave process involving a large-amplitude monochromatic Alfvén pump wave propagating parallel to $B_0$, a spectrum of electro-
195    static ion acoustic waves also at parallel propagation, and a spectrum of Alfven daughter waves at anti-parallel propagation."

– *Page 3, Line 22: Delete linear (saturation is a nonlinear process).*
Reply:
Done.

200    Changes in the manuscript:

– Page 3: Line 28.

– *Page 3, Line 23: Delete nonlinear; it is unnecessary.*

Reply:
205    Done.

Changes in the manuscript:

– Page 3: Line 29.

– *Page 3, Line 33: ... and the lower panels correspond to the end of the simulation (t $\Omega_{cp}$*
210    *=600)."*

Reply:
Done.

Changes in the manuscript:

215     – Page 4: Line 7
       "... and the lower panels correspond to the end of the simulation (t $\Omega_p$ =600)."

  – *Page 6: Insert the definitions of the solid lines and the dashed lines in the caption to Figure 3.*

    Reply:
    Done.

220     Changes in the manuscript:

    – Caption of Fig. 3:
      "... The dashed lines describe the locus $(v_\parallel, v_\perp)$ of the particle velocities
      $v = \sqrt{(v_\parallel - V_{ph})^2 + v_\perp^2}$ where their energy is conserved in the wave frame. Here $V_{ph}$ is
225       the phase speed of the Alfvén wave."

    Other changes in the manuscript:

    – Page: 3, Line 11 to line 15
      "The value of beta parameter 0.01 is set in order to keep the same value of beta in all the
230       1-D, 2-D, and 3-D setups for the purpose of comparison, and moreover, a lower value of
      beta (such as 0.01) is not irrelevant from the solar wind studies. In fact, the solar wind
      plasma originates in the corona and the low-beta plasmas are more representative in the
      inner heliosphere. Therefore, we regard our numerical studies not only for understanding
      the solar wind but also for understanding the solar corona."

235     – Page 4: Line 4 to Line 6
      "The time evolution of the velocity distribution functions is usually helpful to emphasize
      the role of the kinetic regime on the saturation of the instability via particle trapping and
      wave particle interactions."

    – Page 4: Line 13 to Line 16
240       "This is a consequence of the low values for the electron beta ($\beta$=0.01) and ion beta
      ($\beta$=0.01) used in the simulation. At very low electron temperatures, the contribution of
      the electron pressure term to the electric field ($\nabla P_e = \nabla n k_B T_e$) is small and the particle
      density fluctuations $n$ are less efficient in coupling to the electric field fluctuations."

**On heating of solar wind protons by the  *parametric decay* of large amplitude Alfvén waves**

Horia Comişel[1,2], Yasuhiro Nariyuki[3], Yasuhito Narita[4,5], and Uwe Motschmann[1,6]

[1]Institut für Theoretische Physik, Technische Universität Braunschweig, Mendelssohnstr. 3, D-38106 Braunschweig, Germany
[2]Institute for Space Sciences, Atomiştilor 409, P.O. Box MG-23, Bucharest-Măgurele, RO-077125, Romania
[3]Faculty of Human Development, University of Toyama, 3190, Gofuku, Toyama City, Toyama 930-8555, Japan
[4]Space Research Institute, Austrian Academy of Sciences, Schmiedlstr. 6, A-8042 Graz, Austria
[5]Institut für Geophysik und extraterrestrische Physik, Technische Universität Braunschweig, Mendelssohnstr. 3, D-38106 Braunschweig, Germany
[6]Deutsches Zentrum für Luft- und Raumfahrt, Institut für Planetenforschung, Rutherfordstr. 2, D-12489 Berlin, Germany

**Correspondence:** H. Comişel
(h.comisel@tu-braunschweig.de)

**Abstract.**

By means of hybrid simulations, we present a study on  **ion** heating by the field-aligned parametric decay of a monochromatic left-hand polarized Alfvén wave. **The comparison made among different spatial dimensions proves that the three-dimensional simulation exhibits more efficient heating. Plasma is heated parallel to the mean magnetic field by the damping of the ion acoustic waves while being heated perpendicular by the cyclotron resonance and damping of protons by Alfvén daughter waves. In the solar wind context, the antisunward part of the core component of the proton velocity distributions is controlled by the sunward-propagating waves driven by the parametric decay.**

[revised manuscript text omitted]
 0.01 is set in order to keep the same value of beta in all the 1-D, 2-D, and 3-D setups for the purpose of comparison, and moreover, a lower value of beta (such as 0.01) is not irrelevant from the solar wind studies. In fact, the solar wind plasma originates in the corona and the low-beta plasmas are more representative in the inner heliosphere. Therefore, we regard our numerical studies not only for understanding the solar wind but also for understanding the solar corona.** At the initial time, a monochromatic Alfvén pump wave with left-handed circular polarization is launched parallel to the mean magnetic field along the Oz axis of the simulation box. The amplitude of the pump wave is set to 20% of the mean magnetic field magnitude, $\delta B/B_0$=0.2. The wavenumber of the pump wave is $k_0 V_A/\Omega_p = 0.218$, corresponding to a Fourier mode of $m$=10 ($m$=25 for the 1D system). The initial  **fluctuating** magnetic field ($\boldsymbol{B}_\perp$) and bulk velocity ($\boldsymbol{u}_\perp$) satisfies the relation $\boldsymbol{u}_\perp = -k_0/\omega_0 \boldsymbol{B}_\perp$, while the resonant frequency $\omega_0$=0.196 $\Omega_p$, is determined from the dispersion relation $k_0^2 = \omega_0^2/(1 - \omega_0)$ for the left-handed waves, (see e.g., Terasawa et al., 1986). The initial setup of the pump waves is identical for all the 1D, 2D and 3D configurations.

**The parametric decay modeled here is a three-wave process involving a large-amplitude monochromatic Alfvén pump wave propagating parallel to $B_0$, a spectrum of electrostatic ion acoustic waves also at parallel propagation, and a spectrum of Alfvén daughter waves at anti-parallel propagation.** Figure 1 shows the time profiles for the magnetic field energy of the pump wave (labeled by $E^P$) and the backward propagating Alfvén daughter waves (labeled by $E^D$) for the 2D setup (left panel) and the 3D setup (right panel). Overplotted by gray solid lines are the root mean square (rms) density fluctuations ($\rho$) with a maximum at about 300 ion gyroperiods when the linear growing of $E^D$ terminates, i.e., the  saturation of the instability occurs. At the  saturation time when the energy of the daughter wave overtakes the energy of the mother wave, the frequency-wavenumber power spectrum of the magnetic field is shown in the lower panels of Fig. 1. The left-handed pump wave (here we adopted the convention of negative frequency for the left-handed mode) marked with $E^P$ can be seen in the lower-right quadrant ($k > 0$ and $\omega < 0$) while the counter-propagating Alfvén daughter wave ($E^D$) is developed in the lower-left quadrant ($k < 0$ and $\omega < 0$) of the frequency-wavenumber spectrum and has the Fourier mode $m$=-8 ($m$=-23 for the 1D run, not shown here). The waves driven by the decay instability dominate the entire spectrum.

[Figure]

**Figure 1.** Top: Time evolution of the normalized magnetic field energy of the Alfvén pump wave ($E^P$) and the counter propagating Alfvén daughter wave ($E^D$). Overplotted by gray are given the rms density fluctuations $\rho$. Bottom: Power spectrum in the wavenumber frequency domain of the magnetic field determined around time $t\Omega_p \sim 500$.

Qualitatively speaking, the results presented in Fig. 1 are consistent with the earlier MHD studies, proving that the parametric decay responsible for the breaking of the Alfvén pump wave occurs irrespective of the spatial dimensions.

Figure 2 shows the particle distribution functions in the phase space $z - v_\parallel$ at two different stages for the evolution of decay instability in the 2-D and 3-D setup. **The time evolution of the velocity distribution functions is usually helpful to**

5  **emphasize the role of the kinetic regime on the saturation of the instability via particle trapping and wave particle interactions.** The upper panels of Fig. 2 refer to the linear stage of the instability at a time of $t\Omega_p \sim 300$ and the lower panels **correspond to the end of the simulation** ($t\Omega_p = 600$). The number of particles is represented in a color code scale with a minimum defined by the lightest nuance of gray. At the linear stage, the protons are accelerated by the parallel electric field produced by the density fluctuations. The proton phase space $z - v_\parallel$ is close to the result from an earlier study. Matteini et

10  al. (2010a) explain the spatial modulation and the modulation in enhancement of the parallel electric field (Fig. 5 in their paper) due to the broader spectrum of ion acoustic waves excited by the large amplitude Alfvén mother wave. The spatial modulation in Fig. 2 is weaker according to the smaller amplitude pump wave used in our simulation. During the saturation of the instability we do not observe the presence of phase-space vortices leading to the formation of a proton beam. **This is a consequence of the low values for the electron beta ($\beta$=0.01) and ion beta ($\beta$=0.01) used in the simulation. At very low**

15  **electron temperatures, the contribution of the electron pressure term to the electric field ($\nabla P_e = \nabla n k_B T_e$) is small and the particle density fluctuations $n$ are less efficient in coupling to the electric field fluctuations.** The proton phase space at

[Figure]

**Figure 2.** Top: Proton phase space $z - v_\parallel$ at time $t\Omega_p = 300$ close to the linear saturation of the decay instability. Bottom: Proton phase space $z - v_\parallel$ at the final time $t\Omega_p = 600$ of the simulation.

the linear stage of the instability does not differ too much between the 2D and 3D runs. The velocity distribution, in contrast, is different for the 3D system at the nonlinear stage of the instability as one can see in the lower panels of Fig. 2. Particles are smoothly heated and the resonant protons accelerated by the ion acoustic waves are mixed with the thermal core of the distribution.

5    Figure 3 reports a comparison of the proton velocity distribution functions constructed in the $(v_\perp, v_\parallel)$ plane obtained from the one-, two-, and three- dimensional systems **at the initial time** and at the same simulation epochs as in Figure 2. **The velocity distribution function $f$ is computed by counting the number of particles $dN = f(v_\perp, v_\parallel, \Phi)dV$ in the volume element $dV = v_\perp dv_\perp dv_\parallel d\Phi$, and by integrating over the azimuthal $\Phi$ angle. Here, $v_\perp = \sqrt{v_{\perp 1}^2 + v_{\perp 2}^2}$ is the velocity component perpendicular to the mean magnetic field, $v_\parallel$ is the parallel velocity, and the angle $\Phi = \arctan v_{\perp 1}/v_{\perp 2}$ gives the sign of**

10   $v_\perp$**.** The contour lines are shown for fractions of 0.6, 0.3, 0.2, and 0.1 of the maximum phase space density from the inner to the outer part of the distribution functions. **Due to the transversal wave field imposed at the initial condition, the velocity distribution functions are rigid shifted towards the initial bulk velocity $\pm u_\perp$, see e.g., Verscharen and Marsch (2011); Nariyuki (2011). The two symmetrical sets of contour levels with respect to the $v_\perp = 0$ axis are slowly merging with the time evolution and at time t$\Omega_p$=300 there are no remnants of the rigid displacement observed at the initial time.** Similar

with Fig. 2, the velocity distribution functions  do not clearly differ close to the linear stage of the decay instability at time $t\Omega_{\mathrm{p}}$=300. The dashed lines represent arcs of circles with centers localized along the $v_{\parallel}$ axis at the position corresponding to the phase velocities of Alfvén pump wave ($V_{ph} = \omega_0/k_0 V_{\mathrm{A}} \approx$-0.9) and Alfvén daughter wave ($V_{ph} = \omega/k V_{\mathrm{A}} \approx$+0.9), respectively. They describe the particle velocity in the wave frame, $v = \sqrt{(v_{\parallel} - V_{ph})^2 + v_{\perp}^2}$. According to pitch angle scattering model, the particle energy, E=m$v^2$/2, is conserved in this reference system.

**Although there are no obvious differences in the three analyzed setups, one can notice a distinct trend of evolution for the distributions between the intermediate ($t\Omega_p$=300) and the final time of simulation ($t\Omega_p$=600). At the final time, the contour levels in the 3D simulation are moderately enlarging both in the parallel and perpendicular directions by following the contour levels of energy conservation driven by the pitch-angle scattering of protons. The contour levels in 1D and 2D runs, in contrast, are developing mainly towards parallel direction while their initial perpendicular displacement is removing with time elapsing. The more efficient heating of plasma in the 3D system is consistent with the time evolution of the ion temperature shown in Figure 4. The particles experience a similar parallel heating while the perpendicular temperature achieved in the 3D system (solid line) dominates by a factor of two or more the corresponding values obtained in the 1D (dotted line) and 2D (dashed line) simulations.**

**These results suggest that the damping of the ion sound waves excited by the field aligned parametric decay is the main mechanism of plasma heating in the 1D and 2D systems. In the 3D system, the protons are also heated in the perpendicular direction by the cyclotron damping of waves. The ions are perpendicular scattered by the field aligned and the oblique developed Alfvén daughter waves.**

**Figure 1 shows that the amplitude of the anti-parallel propagating Alfvén daughter wave decreases with increasing spatial dimension, while the level of the density fluctuation is similar. The power spectrum $\delta B^2(k_{\parallel})$ is obtained by the Fourier transformation of the averaged magnetic field, $\delta\tilde{B}(r_{\parallel}) = \int \delta B(r_{\perp 1}, r_{\perp 2}, r_{\parallel}) dr_{\perp 1} dr_{\perp 2}$, in the assumption of strictly parallel wave propagation. Any deviation from the parallel direction will conduct to a reduction of the $E^D$ amplitude of the daughter mode. A slight oblicity (several degrees) of the daughter wave mode is noticed in the 3D system but the reason remains unclear within the simulation work here and needs further investigations. The pitch-angle scattering and the perpendicular temperature increase observed in the time evolutions of the velocity distribution functions and temperatures, respectively, suggest that the Alfvén daughter waves are in cyclotron resonance with protons and the wave-particle interaction could explain the deviation in the propagation angle and the stronger damping of the daughter waves in the 3D system. A detailed spectral analyzing of the oblique wave modes developed in the decay process based on the 2D reduced magnetic field spectrum will be subject for a further study.**

**3    Conclusion**

We studied the proton acceleration and heating driven by the parametric decay of a large amplitude Alfvén wave in the linear and nonlinear stage of the instability in a multidimensional system. By comparing the wave modes and proton velocity distribution functions **in 1D, 2D, and 3D systems, we conclude that the plasma is heated more efficient in the 3D system,**

[Figure]

**Figure 3.** Proton distribution functions represented as contour levels (solid line) determined in the plane ($v_\perp$,$v_\parallel$) **at times $t\Omega_p$=0 (a), $t\Omega_p$=300 (b), and $t\Omega_p$=600 (c). The dashed lines describe the locus ($v_\parallel$, $v_\perp$) of the particle velocities $v = \sqrt{(v_\parallel - V_{ph})^2 + v_\perp^2}$ where their energy is conserved in the wave frame. Here $V_{ph}$ is the phase speed of the Alfvén wave.**

**thus proving that the 3D simulations yield better results than the corresponding 1D and 2D results. Parallel heating of plasma is provided by the damping of ion sound waves while perpendicular heating is given by the perpendicular scattering of protons by the field aligned and the oblique developed Alfvén daughter waves. The pitch-angle scattering is the mechanism able to describe the perpendicular broadening observed in the particle velocity distribution functions.**

5   In the 3D system, the conditions for resonant cyclotron interactions of ions with the left-handed Alfvén waves are apparently

[Figure]

**Figure 4.** Time evolution of the parallel ($T_\parallel$) and perpendicular ($T_\perp$) proton temperatures for the 1-D (dotted line), 2-D (dashed line), and 3-D (solid line) configurations. The temperatures are normalized to their initial values.

[revised manuscript text omitted]

---

## Author Comment (AC2) · 16 Apr 2018

**Reply to referee comments**

On heating of solar wind protons by the breaking of large amplitude Alfvén waves

Manuscript ID: angeo-2018-14 (AngeoComm)
H. Comişel, Y. Nariyuki, Y. Narita, and U. Motschmann

Thank you very much for reading the manuscript and raising helpful comments and suggestions.

[Figure]

- *The manuscript #angeo-2018-14 by ComiÂÿsel et al. discusses heating of protons during the course of nonlinear evolution of a large amplitude Alfven wave in the solar wind. The author used a hybrid code to conduct 1D, 2D, and 3D simulations all started with a circularly polarized pump Alfven wave imposed at the initial condition. The simulation results showed that an efficient proton heating in the perpendicular direction occurs in 3D, but not in 1D and 2D. I think the finding itself is interesting and the paper is potentially worth for publication. However, the authors' discussion does not explain how the discrepancy arises between the different simulations. Also, there is a plenty of room for improvement in the quality of presentation. Therefore, I believe that substantial revisions need to be made on the manuscript before the publication.*

*Major Issues*

*The simulation results suggested that pitch-angle scattering by both the pump and daughter Alfven waves are more important than the heating in the parallel direction via damping of ion sound waves in 3D. In contrast, 1D and 2D results suggested that the pitch-angle scattering are less important compared to 3D. It is not entirely clear how the difference arises. If you look at Figure 1, you can clearly see that the ion sound wave amplitudes do not change much between 2D and 3D. On the other hand, the amplitude of the daughter Alfven wave in 3D is much less than that in 2D. Apparently, this contradicts with the behavior seen in the distribution function. The 3D simulation presented in the paper is quite large and should contain a lot of information. My impression is that the authors did not make use of the benefits of the large-scale simulation. I would encourage the authors to conduct more detailed analysis and draw more physically sound and grounded conclusions based on the data.*

Reply:

Figure 1 shows indeed that the amplitude of the anti-parallel propagating Alfvén daughter wave decreases with increasing spatial dimension, while the level of the density fluctuation is similar. The power spectrum $\delta \vec{B}^2(k_\parallel)$ is obtained by the Fourier transformation of the averaged magnetic field,

$$\delta \tilde{\vec{B}}(r_\parallel) = \int \delta \vec{B}(r_{\perp 1}, r_{\perp 2}, r_\parallel) dr_{\perp 1} dr_{\perp 2},$$

in the assumption of strictly parallel wave propagation. Any deviation from the parallel direction will conduct to a reduction of the $E^D$ amplitude of the daughter mode. A slight obliquity (several degrees) of the daughter wave mode is noticed in the 3D system but the reason remains unclear within the simulation work here and needs further investigations. The pitch-angle scattering and the perpendicular temperature increase observed in the time evolutions of the velocity distribution functions and temperatures, respectively, suggest that the Alfvén daughter waves are in cyclotron resonance with protons and the wave-particle interaction could explain the deviation in the propagation angle and the stronger damping of the daughter waves in the 3D system. A detailed spectral analyzing of the oblique wave modes developed in the decay process based on the 2D reduced magnetic field spectrum will be subject for a further study.

The velocity distribution functions have been updated (please see our answer concerning Fig. 3 below). Although there are no obvious differences in the three analyzed setups, one can notice a distinct trend of evolution for the distributions between the intermediate ($t\Omega_p$=300) and the final time of simulation ($t\Omega_p$=600). At the final time, the contour levels in the 3D simulation are moderately enlarging both in the parallel and perpendicular directions by following the contour levels of energy conservation driven by the pitch-angle scattering of protons. The contour

 **ANGEOD**

Interactive
comment

levels in 1D and 2D runs, in contrast, are developing mainly towards parallel direction while their initial perpendicular displacement is removing with time elapsing. The more efficient heating of plasma in the 3D system is consistent with the time evolution of the ion temperature shown in the new added Figure 4. The particles experience a similar parallel heating while the perpendicular temperature achieved in the 3D system (solid line) dominates by a factor of two or more the corresponding values obtained in the 1D (dotted line) and 2D (dashed line) simulations.

These results suggest that the damping of the ion sound waves excited by the field aligned parametric decay is the main mechanism of plasma heating in the 1D and 2D systems. In the 3D system, the protons are also heated in the perpendicular direction by the cyclotron damping of waves. The ions are perpendicular scattered by the field aligned and the oblique developed Alfvén daughter waves.

Changes in the manuscript:

- Former manuscript, Page 6: Line 10 to Line Line 14 - deleted.
- Page 6: Line 19 to Line 29
  "Figure 1 shows that the amplitude of the anti-parallel propagating Alfvén daughter wave decreases with increasing spatial dimension, while the level of the density fluctuation is similar. The power spectrum $\delta\vec{B}^2(k_\parallel)$ is obtained by the Fourier transformation of the averaged magnetic field, $\delta\tilde{\vec{B}}(r_\parallel) = \int \delta\vec{B}(r_{\perp 1}, r_{\perp 2}, r_\parallel) dr_{\perp 1} dr_{\perp 2}$, in the assumption of strictly parallel wave propagation. Any deviation from the parallel direction will conduct to a reduction of the $E^D$ amplitude of the daughter mode. A slight obliquity (several degrees) of the daughter wave mode is noticed in the 3D system but the reason remains unclear within the simulation work here and needs further in-

 **ANGEOD**

Interactive
comment

 **ANGEOD**

Interactive
comment

levels in 1D and 2D runs, in contrast, are developing mainly towards parallel direction while their initial perpendicular displacement is removing with time elapsing. The more efficient heating of plasma in the 3D system is consistent with the time evolution of the ion temperature shown in the new added Figure 4. The particles experience a similar parallel heating while the perpendicular temperature achieved in the 3D system (solid line) dominates by a factor of two or more the corresponding values obtained in the 1D (dotted line) and 2D (dashed line) simulations.

These results suggest that the damping of the ion sound waves excited by the field aligned parametric decay is the main mechanism of plasma heating in the 1D and 2D systems. In the 3D system, the protons are also heated in the perpendicular direction by the cyclotron damping of waves. The ions are perpendicular scattered by the field aligned and the oblique developed Alfvén daughter waves.

Changes in the manuscript:

- Former manuscript, Page 6: Line 10 to Line Line 14 - deleted.
- Page 6: Line 19 to Line 29
  "Figure 1 shows that the amplitude of the anti-parallel propagating Alfvén daughter wave decreases with increasing spatial dimension, while the level of the density fluctuation is similar. The power spectrum $\delta\vec{B}^2(k_\parallel)$ is obtained by the Fourier transformation of the averaged magnetic field, $\delta\tilde{\vec{B}}(r_\parallel) = \int \delta\vec{B}(r_{\perp 1}, r_{\perp 2}, r_\parallel) dr_{\perp 1} dr_{\perp 2}$, in the assumption of strictly parallel wave propagation. Any deviation from the parallel direction will conduct to a reduction of the $E^D$ amplitude of the daughter mode. A slight obliquity (several degrees) of the daughter wave mode is noticed in the 3D system but the reason remains unclear within the simulation work here and needs further in-

vestigations. The pitch-angle scattering and the perpendicular temperature increase observed in the time evolutions of the velocity distribution functions and temperatures, respectively, suggest that the Alfvén daughter waves are in cyclotron resonance with protons and the wave-particle interaction could explain the deviation in the propagation angle and the stronger damping of the daughter waves in the 3D system. A detailed spectral analyzing of the oblique wave modes developed in the decay process based on the 2D reduced magnetic field spectrum will be subject for a further study."

– Page 6: Line 6 to Line 18
"Although there are no obvious differences in the three analyzed setups, one can notice a distinct trend of evolution for the distributions between the intermediate ($t\Omega_p$=300) and the final time of simulation ($t\Omega_p$=600). At the final time, the contour levels in the 3D simulation are moderately enlarging both in the parallel and perpendicular directions by following the contour levels of energy conservation driven by the pitch-angle scattering of protons. The contour levels in 1D and 2D runs, in contrast, are developing mainly towards parallel direction while their initial perpendicular displacement is removing with time elapsing. The more efficient heating of plasma in the 3D system is consistent with the time evolution of the ion temperature shown in Figure 4. The particles experience a similar parallel heating while the perpendicular temperature achieved in the 3D system (solid line) dominates by a factor of two or more the corresponding values obtained in the 1D (dotted line) and 2D (dashed line) simulations.

These results suggest that the damping of the ion sound waves excited by the field aligned parametric decay is the main mechanism of plasma heating in the 1D and 2D systems. In the 3D system, the protons are also heated in the perpendicular direction by the cyclotron damping of waves. The ions are perpendicular scattered by the field aligned and the oblique developed Alfvén daughter waves."

• *The authors used beta = 0.01, which is very small in comparison with the solar wind at 1 AU. Please consider to state the motivation for adopting this particular value. Also, the author may think it better to include discussion on the dependence of the plasma beta.*

Reply:

We agree and accept the referee's criticism. It is true that we are aiming at the simulations for a higher value of beta (ideally a beta value of unity for the solar wind at 1 AU) but the numerical costs become increasingly higher at higher values of beta and in a three-dimensional setup. The reason for this is the fact that a considerably large number of particles needs to be used for high-beta plasmas and also the number of cells or mesh grids increases in three dimensions.

The value of beta value 0.01 is set in order to keep the same value of beta in all the 1-D, 2-D, and 3-D setups for the purpose of comparison, and moreover, a lower value of beta (such as 0.01) is not irrelevant from the solar wind studies. In fact, the solar wind plasma originates in the corona and the low-beta plasmas are more representative in the inner heliosphere. Therefore, we regard our numerical studies not only for understanding the solar wind but also for understanding the solar corona.

Changes in the manuscript:

– Page 3: Line 11 to Line 15
  "The value of beta parameter 0.01 is set in order to keep the same value of beta in all the 1-D, 2-D, and 3-D setups for the purpose of comparison, and moreover, a lower value of beta (such as 0.01) is not irrelevant from the solar wind studies. In fact, the solar wind plasma originates in the corona and the low-beta plasmas are more representative in the inner heliosphere.

Therefore, we regard our numerical studies not only for understanding the solar wind but also for understanding the solar corona."

- *P.2, L. 23: "the conditions of beta plasmas" does not make sense.*

  Reply:
  We specify now low-beta plasmas.

  Changes in the manuscript:

  - Page 2: Line 23
    " ... the conditions of low-beta plasmas".

- *P.3, L. 29: The sentence ending with "due to most probably the small value of electron beta used in simulation" needs more clarification.*

  Reply:
  We delete "most probably" and add the sentences below.

  Changes in the manuscript:

  - Page: 4 Line 13 to Line 16
    "This is a consequence of the low values for the electron beta ($\beta$=0.01)

and ion beta ($\beta$=0.01) used in the simulation. At low electron temperatures, the contribution of the electron pressure term to the electric field ($\nabla P_e = \nabla n k_B T_e$) is small and the particle density fluctuations $n$ are less efficient in coupling to the electric field fluctuations.".

- *P.4, L. 25: What the authors meant by the sentence "We see that these arcs..." is not clear. I can guess what you wanted to mention, but in general, it is not a good idea to let the readers guess the meanings.*

Reply:
This sentence and the following ones (Page 5, Line 16 to Page 6, Line 9 in the former manuscript) have been deleted. Instead, we introduced a new comment discussed above (Page 6, Line 6 to Line 18).

- *Figure 2: Are the color scales linear or logarithmic? Are they the same or different between the four panels? It is difficult for me to see behaviors in the low-energy part of the distribution function. Is this the authors' intention?*

Reply:
The color code is given in logarithmic scales. The color code bars are now added to each panel in new Figure 2. The purpose of using the color code representation is to identify the occurrence of vortices in the velocity phase space.

Changes in the manuscript:

- Figure 2 is updated.

- Page: 4 Line 4 to Line 6
  "The time evolution of the velocity distribution functions is usually helpful to emphasize the role of the kinetic regime on the saturation of the instability via particle trapping and wave particle interactions."

- *Figure 3: It would be better to add a description for the dashed lines in the caption. The authors' definition of $v_perp$ should never become negative. You need a more clear description what the distribution functions in Figure 3 represent.*

Reply:
Done.
Figure 3 has been updated by including a missing term $(1/v_\perp)$ used to properly compute the velocity distribution functions in cylindrical coordinates. The velocity distribution function $f$ is computed by counting the number of particles $dN = f(v_\perp, v_\parallel, \Phi)dV$ in the volume element $dV = v_\perp dv_\perp dv_\parallel d\Phi$, and by integrating over the azimuthal $\Phi$ angle. Here, $v_\perp = \sqrt{v_{\perp 1}^2 + v_{\perp 2}^2}$ is the velocity component perpendicular to the mean magnetic field, $v_\parallel$ is the parallel velocity, and the angle $\Phi = \arctan v_{\perp 1}/v_{\perp 2}$ gives the sign of $v_\perp$. The updated plots are slightly different from those shown in former Fig. 3. The distribution functions at the initial time are also introduced.

Changes in the manuscript:

- The caption of Fig. 3
  "... The dashed lines describe the locus $(v_\parallel, v_\perp)$ of the particle velocities

$v = \sqrt{(v_\parallel - V_{ph})^2 + v_\perp^2}$ where their energy is conserved in the wave frame. Here $V_{ph}$ is the phase speed of the Alfvén wave."

– Page 5: Line 7 to Line 9
"The velocity distribution function $f$ is computed by counting the number of particles $dN = f(v_\perp, v_\parallel, \Phi)dV$ in the volume element $dV = v_\perp dv_\perp dv_\parallel d\Phi$, and by integrating over the azimuthal $\Phi$ angle. Here, $v_\perp = \sqrt{v_{\perp 1}^2 + v_{\perp 2}^2}$ is the velocity component perpendicular to the mean magnetic field, $v_\parallel$ is the parallel velocity, and the angle $\Phi = \arctan v_{\perp 1}/v_{\perp 2}$ gives the sign of $v_\perp$ in Fig. 3."

– Page 5: Line 11 to Line 14
"Due to the transversal wave field imposed at the initial condition, the velocity distribution functions are rigid shifted towards the initial bulk velocity $\pm u_\perp$, see e.g., Verscharen2011, Nariyuki2011. The two symmetrical sets of contour levels with respect to the $v_\perp = 0$ axis are slowly merging with the time evolution and at time t$\Omega_p$=300 there are no remnants of the rigid displacement observed at the initial time."

– Former manuscript, Page 5: Line 1 to Line 3 - deleted.

Other changes in the manuscript:

– Title of the paper: Instead "breaking of ..." we introduced "parametric decay of ..." in order to avoid any confusion.
– Abstract Page 1: Line 3 to Line 7
"The comparison made among different spatial dimensions proves that the three-dimensional simulation exhibits more efficient heating. Plasma is

heated parallel to the mean magnetic field by the damping of the ion acoustic waves while being heated perpendicular by the cyclotron resonance and damping of protons by Alfvén daughter waves. In the solar wind context, the antisunward part of the core component of the proton velocity distributions is controlled by the sunward-propagating waves driven by the parametric decay."

– Page 6: Line 32 to Page 7, Line 4

"By comparing the wave modes and proton velocity distribution functions in 1D, 2D, and 3D systems, we conclude that the plasma is heated more efficient in the 3D system, thus proving that the 3D simulations yield better results than the corresponding 1D and 2D results. Parallel heating of plasma is provided by the damping of ion sound waves while perpendicular heating is given by the perpendicular scattering of protons by the field aligned and the oblique developed Alfvén daughter waves. The pitch-angle scattering is the mechanism able to describe the perpendicular broadening observed in the particle velocity distribution functions."

---

## Author Response (AR2)

**Reply to referee comments**

On heating of solar wind protons by the parametric decay of large amplitude
Alfvén waves

Manuscript ID: angeo-2018-14 (AngeoComm)

H. Comişel, Y. Nariyuki, Y. Narita, and U. Motschmann

*I thank the authors for their positive responses to my comments. The revised manuscript is definitely improved, but I am sorry to report that I have problems with some of the revised material. Therefore, I feel it is necessary to request another revision of the manuscript.*

*Abstract: Replace the second and third sentences with Three simulations with the same physics are carried out with spatial variations in one dimension, two dimensions, and three dimensions (3D); comparison shows that the 3D simulation yields the strongest proton heating. The protons are heated parallel...*

*Page 1, line 3: Replace The comparison made among different spatial dimensions proves with Comparison is made among one-dimensional, two-dimensional, and three-dimensional simulations, showing that.*

*Page 1, line 3: Replace proves with demonstrates.*

*Page 1, line 4: Replace more efficient with the strongest proton.*

*Page 1, line 4: Replace Plasma is with Protons are.*

*Page 1, line 13 and several other locations: Replace plasma heating with proton heating. Plasma heating implies both ions and electrons are heated, whereas only proton heating is described here.*

*Page 2, line 2: Replace outwards with away from.*

*Page 2, line 11: Replace plasma with proton.*

*Page 2, line 18: Replace prove with show.*

*Page 2, line 22: Replace plasmas with ions.*

*Page 2, line 27: Replace plasma with proton.*

*Page 3, line 2: The properties of the 1D and 2D runs should be clarified: In the one-dimensional box spatial variations are allowed only in directions parallel and antiparallel to the background magnetic field, whereas in the two-dimensional simulation spatial variations are allowed in both the parallel/antiparallel direction and one perpendicular direction.*

*Page 3, lines 13-15: Delete is not irrelevant from the solar wind studies. In fact..the solar corona and simply say is relevant for the solar corona and inner heliosphere.*

*Page 3, line 21: Replace waves with wave.*

*Page 5 lines 12-14: The terms rigid shifted and rigid displacement are unclear. The t=0 distributions in the top row suggest a donut shape; perhaps that term might be used. Also, Figure 1 of Verscharen and Marsch (2011) is very different from the top row of Figure 3. Please explain.*

50

*Page 6, line 11: Replace The more efficient heating of plasma with The stronger heating of protons.*

*Page 6, line 16: Replace plasma with proton.*

55 *Page 6, lines 28-29: Delete the sentence beginning A detailed spectral analyzing... Promises of future work are inappropriate in a scientific manuscript.*

*Page 6, line 33: Replace more efficient with most strongly.*

60 *Page 7, line 2: Replace plasma with protons.*

==================================================================
Reply:

We thank the reviewer for reading the revised manuscript and raising helpful suggestions.
65 The actual manuscript has a major revision mainly based on the analysis of the 3-D simulation results. In this revised manuscript, the above suggestions have been considered. Here we have a comment regarding the following question raised by the reviewer.

*Page 5 lines 12-14: The terms rigid shifted and rigid displacement are unclear. The t=0 distributions*
70 *in the top row suggest a donut shape; perhaps that term might be used. Also, Figure 1 of Verscharen and Marsch (2011) is very different from the top row of Figure 3. Please explain.*
Reply:

This is true. Our velocity distributions at time $t\Omega_p$=0 are similar with the particle-in-cell simulation
75 discussed by Sakai et al., 2005 (Fig. 10). Figure 1 of Verscharen and Marsch (2011) refers to the effect of the wave field of force resulting in the apparent temperature anisotropy at an intermediate time ($t\Omega_p > 0$ ). However, the different time evolution in our three simulation setups suggests that the apparent temperature anisotropy does not significantly influence the proton heating reported at the latest time of simulation.

80

Sakai , J.I., Yamamura, W., Saito, S., Washimi, H., Tsiklauri, D., and Vekstein, G.: Particle simulation of plasma heating by a large-amplitude shear Alfvén wave through its transverse modulation in collisionless plasmas, New J. Phys., 7, 233, https://doi.org/10.1088/1367-2630/7/1/233, 2005.

85 Changes in the manuscript:
Page 9, Row: 17-19:
" We also mention that starting from similar velocity distribution functions distorted by the initial wave fields, their different time evolution in the three simulation setups suggests that the apparent temperature anisotropy does not significantly influence the proton heating reported at the latest time
90 of simulation."

---

## Author Response (AR3)

**Reply to referee comments**

On heating of solar wind protons by the parametric decay of large amplitude Alfvén waves

Manuscript ID: angeo-2018-14 (AngeoComm)

H. Comişel, Y. Nariyuki, Y. Narita, and U. Motschmann

We thank the reviewer for reading the manuscript and raising helpful questions and suggestions. Here we give our reply. The changes in the manuscript are marked by bold characters.

The manuscript presents a 3D hybrid simulation of the parametric decay of an Alfven cyclotron wave
in a low beta plasma (0.01). Most of these 3D simulation results seems to be similar to previous 1D and 2D results in many respects. The exception is much larger perpendicular heating in 3D compared to 1D and 2D results. This findings is very interesting and makes the manuscript relevant for AG. However, no attempt is made to find out why such a big difference exists between 3D and 2D (1D geometry is clearly very different). Furthermore, there are some issues with the numerical made and the manuscript contains are expressed as a second s model and the manuscript contains some erroneous claims. Consequently, the manuscript needs a major revision.

The presented results have numerical issues: the used spatial resolution  $(=1d_i)$  is insufficient to 20 resolve the proton gyroradius  $(rho_i \ 0.1 \ d_i)$  so that the presented results are likely comparable to multi-fluid ones. This should be discussed. It is also necessary to include all the relevant physical and numerical parameters used for the (1D, 2D, and 3D) simulations.

Reply:

The used spatial resolution for the field quantities (magnetic field, electric field, and velocity moments) is indeed close to the ion inertial length (the updated simulation has  $0.5 d_i$  spatial resolution along the Oz axis and  $1 d_i$  along the perpendicular axes) and the smaller spatial gradients cannot be resolved. The magnetic field within a numerical cell is overall homogeneous with the linear interpolations at the particle position between mesh points or due to the wave magnetic field. Thus, the perpendicular projection of the proton motion is nearly a circle and this circular gyration is resolved by about 100 time steps. Gradients become important over about 10 gyroradii or more and not just over one gyration.

Changes in the manuscript:

Page:3, Line:14 to Page:4, Line:2

"The used spatial resolution for the field quantities (magnetic field, electric field, and velocity moments) is close to the ion inertial length and the proton gyroradius ( $\rho_i \sim 0.1 d_i$ ) or smaller spatial gradients cannot be resolved. The magnetic field within a numerical cell is overall homogeneous with the linear interpolations at the particle position between mesh points or due to the wave magnetic field. Thus, the perpendicular projection of the proton motion is nearly a circle and this circular gyration is resolved by about 100 time steps. Gradients become important over about 10 gyroradii and not just over one gyration."

Page: 9, Line: 21-22

"... while all the other physical and numerical parameters are basically maintained the same."

**Fig. A.1.** Normalized growth rates (in the same units as Viñas and Goldstein 1991) for the field-aligned (solid line) and oblique (10 deg - dotted,20 deg - dashed) decay instability determined for a left-handed polarized Alfvén wave at  $\beta_{fluid} = 0.02$ .

"... the plasma heating by the Alfvn wave can occur through generation and steepening of magnetosonic waves." The meaning of this is not clear - do you mean that the heating goes through secondary magnetosonic waves?

Reply:

No, here the discussion refers to the ion acoustic waves.

Changes in the manuscript:

Page:2, Line:13

"... the plasma heating by the Alfvén wave can occur through generation and steepening of ion acoustic waves."

"In earlier studies concerning the parametric decay in low beta plasmas (e.g., Del Zanna et al., 2001), at the non-linear saturation time, the cross-helicity c becomes zero, and the wave energy is assumed to be equally shared by the forward-propagating Alfvn pump wave and the backward-propagating Alfvn daughter waves." At least in the case of Del Zanna et al. (2001) this claim is false. The areas helicity tends to change from positive to propagating areas helicity values for low beta

*false. The cross-helicity tends to change from positive to negative cross-helicity values for low beta simulations.*

Reply:

We accept the referee's criticism and change the sentence accordingly.

Changes in the manuscript:

Page:5, Line:14 to Page:6, Line:1

"In earlier studies concerning the parametric decay (e.g., Del Zanna et al. 2001), the cross-helicity of the waves tends to change from positive to negative cross-helicity values for low beta simulations."

"The growth rates for the moderate oblique daughter waves scales by cos theta, accordingly to former studies (e.g., Del Zanna et al., 2001; Matteini et al., 2010a), where theta is the angle of propagation with respect to the mean magnetic field." This is wrong. Del Zanna et al and Matteini et al investigated the dependence of the growth rate on the angle between the magnetic field B and the wave vector of the mother wave  $k_0$ . The dependence on the angle between B and the wave vector of the daughter wave  $k_1$  was numerically studied by Vinas and Goldstein (1991); as far as this referees knows there are no analytical or numerically fitted formula for this.

Reply:

- 85 Yes, this is true. Del Zanna et al. and Matteini et al. investigated the dependence of the growth rate on the angle  $\theta$  between the magnetic field B and the pump wave vector. They found a scaling with  $\cos \theta$  which suggests that the decay process is basically controlled by the field-aligned projection of the pump wave amplitude. Our statement was given in this context.
- The oblique decay instabilities discussed by Viñas and Goldstein (1991) have smaller growth rates at larger propagation angles according to the expectations, and certainly there is no analytical or numerically fitted formula describing the dependence on the propagation angle of the daughter wave. On the other hand, Fig. 3 in Viñas and Goldstein 1991b shows that there is a trend to eliminate/reduce the differences between the growth rates of the oblique and field-aligned decay instabilities while plasma beta values are decreasing to lower values. Following the cited authors' recipe, our calcula-
- 95 tions shown in Fig. A.1 indeed reveal that the growth rates of the decay instability at propagation angles of 0, 10, and 20 deg by using our smaller beta value (equivalent to  $\beta_{fluid} = 0.02$ ) have the same tendency as the proposal by Viñas and Goldstein (1991). This means that the obliquely-propagating waves play a more important role in the dynamics of the heating process in low beta plasmas than the 1-D field-aligned pump case. Furthermore, plasma beta parameter  $\beta$  could be the key factor in
- 100 controlling the perpendicular ion heating by the parametric decay instability in space plasmas.

Changes in the manuscript: The former sentence was replaced by the following comment. Page:5, Line:1-6

"The growth rates for the moderate oblique daughter waves have close values to the parallel daughter mode. Viñas and Goldstein 1991b show that there is a trend to eliminate/reduce the differences between the growth rates of the oblique and field-aligned decay instabilities while plasma beta values are decreasing to lower values. Following Viñas and Goldstein 1991 analytical treatment, our calculations indeed reveal that the oblique growth rates (at propagation angles of 10 and 20 deg)

have close values with the field-aligned growth rate for  $\beta$ =0.01 thus proving the predicted tendency of merging at low beta values."

Page: 11, Line:18-20

"One may expect that plasma beta parameter  $\beta$  could be the key factor in controlling the perpendicular ion heating driven by the parametric decay instability in space plasmas."

*Fig. 3: The distribution functions clearly includes the Alfven wave velocity component. It would be better to suppress this.*

Reply:

- 120 The wave effect on the velocity distribution function can be described by a model distribution function which is equivalent to a Maxwellian distribution shifted by the mean fluid-velocity of the particles associated with the local magnetic field (e.g., Verscharen and Marsch 2011, Nariyuki 2011). In many numerical simulations (e.g., Markovskii et al., 2009) the particle velocity distribution functions are initiated by using such a modified Maxwellian distribution. Here we show that the wave
- 125 field effect due to the pump wave is much less important than the wave field effect driven by the wave decay and oscillation of the daughter waves at later times. The "shift" term added in the Maxwellian distribution functions can be roughly evaluated by determining the ratio between the kinetic bulk energy ( $E_{kin} \sim u_{\perp}^2$ ) and the thermal energy ( $\epsilon_{th} \sim v_{th\perp}^2, v_{th}$  - thermal velocity) for the transverse components with regard to the mean magnetic field. Fig. A.2 shows the time evolution of the term
- 130  $u_{\perp}^2/v_{th\perp}^2$  normalized to its value at time  $t\Omega_p=0$ . As one can see,  $u_{\perp}^2/v_{th\perp}^2$  starts to increase at time  $t\Omega_p > 150$  when the kinetic energy  $E_{kin}$  has more input from the wave decay and oscillation of

Fig. A.2. Time evolution of the ratio between the kinetic and thermal energies normalized to their value at time  $t\Omega_p=0$ .

the daughter waves. Therefore, the suppression of the Alfvén pump wave should change the deformed distribution function shown in Fig. 3 at time  $t\Omega_p=0$  but it should have a minor effect at times  $t\Omega_p=300$  and  $t\Omega_p=600$ .

Changes in the manuscript:

Page: 7, Line:11 to Page:8, Line:6

"The wave effect on the velocity distribution function can be described by a model distribution function which is equivalent to a Maxwellian distribution shifted by the mean fluid-velocity of the particles associated with the local magnetic field (e.g., Verscharen and Marsch 2011, Nariyuki 2011). In many numerical simulations, e.g., Markovskii et al., 2009, such a modified Maxwellian distribution is used at the initial setup to proper account for the motion of particle in the wave field. The wave field effect on the distribution function by applying the pump wave is however much less important on the distribution functions evaluated at later times because the ratio of the bulk and thermal ener- gies describing the velocity shift of the Maxwellian distribution becomes dominant by considering the wave decay and oscillation of the daughter waves."

What is the definition of the temperatures in Fig. 5? Does it include the Alfven wave velocity component as in the case of Fig. 3? If yes it would be better to suppress this. Here we are interested in temperatures and not in effective temperatures.

Reply:

The temperatures in Fig. 5 are determined by computing the thermal velocities (as the second-order velocity moment) obtained by subtracting the bulk velocity from the full particle velocities according to the definition of kinetic temperature,

$$T = \frac{m}{3k_Bn} \int (\mathbf{v}_p - \mathbf{u})^2 f(\mathbf{v}_p) d^3 \mathbf{v}_p$$

where *m* is the mass of the ions, *n* the mean density, **u** the mean bulk velocity,  $\mathbf{v}_p$  the individual particles velocity and  $f(\mathbf{v}_p)$  the velocity distribution functions (see e.g., Baumjohann, W., Treumann, R.A., 1999, Basic Space Plasma Physics, Imperial College Press London; Mueller J.: A.I.K.E.F.: An adaptive hybrid model with application to fossil fields at Titan and Mercury's double magnetopause", adaptive hybrid model with application to fossil fields at Titan and Mercury's double magnetopa PhD Thesis, Technische Universitaet Braunschweig, Pg. 59, ISBN 78-3-942171-63-2, 2012). Changes in the manuscript: Page:10, Line:3-5

"The temperatures are determined by computing the thermal velocities (as the second-order velocity moment) obtained by subtracting the bulk velocity from the full particle velocities according to the definition of kinetic temperature."

With respect to the difference between 2D and 3D results: at least some quantification of the difference between these results (spectrum/amplitude of oblique modes) is necessary.

**Reply:**

A comparison between the 2D and 3D results regarding the spectrum/amplitudes of the oblique modes is shown in the Fig. A.3. The time evolution of the field aligned pump mode  $\delta B(10,0)$ , the field aligned daughter mode  $\delta B(-8,0)$  and the obliquely-propagating daughter mode  $\delta B(-8,8)$  are shown in the top panels of Fig. A.3. The field-aligned density mode  $\delta \rho(18,0)$  and the perpendicular projection  $\delta \rho(0,8)$  of the obliquely-propagating mode  $\delta \rho(18,8)$  are presented in the bottom panels.

The Alfven daughter modes are normalized to the amplitude at the pump mode  $\delta B(10,0)$ .

At the linear stage of the instability growing, the Alfvén oblique mode is about 3 times weaker 180 in amplitude than the field-aligned daughter mode in the 2D setup. In the 3D system the strengths of the oblique and field-aligned daughter modes are nearly the same. Therefore particles are scattered more efficiently by the oblique-propagating waves in the 3D geometry. On the other hand, the perpendicular projection of the obliquely-propagating density mode  $\delta\rho(18,8)$  is more unstable and grows faster than its counterpart from the 2-D setup.

Changes in the manuscript:

The following sentence in the former manuscript was deleted.

"We noticed that the amplitude of the parallel-propagating Alfvén daughter waves decreases with increasing spatial dimension (not shown here) thus proving the efficiency of obliquely-propagating daughter waves prominently developed in the 3-D setup in perpendicularly heating the protons."

New comments are introduced: Page:10, Line:10-15

"With respect to this result we have done a quantification of the differences between the results of the
2-D and 3-D setups in what means the amplitude and slope of the oblique modes. At the linear stage of the instability growing, the Alfvén oblique mode (-8,8) shown in Fig. 2 has close amplitude and slope as the field-aligned daughter mode (-8,0). The correspondent mode in the 2-D setup is about 3 times weaker in amplitude than the field-aligned daughter mode obtained in this configuration. The perpendicular projection of the obliquely-propagating density mode (18,8) is more unstable and grows faster than its counterpart from the 2-D setup. "

*Vinas and Goldstein (1991) show that more channels of parametric instability exist when allowing for oblique daughter waves. Are some of them particularly enhanced in 3D with respect to 2D?*

Reply:

The 3D case opens new channels of the parametric instabilities in the sense that there is a larger population of obliquely propagating waves due to a degree of freedom in the azimuthal direction around the mean magnetic field. As the oblique propagating waves play an important role in the heating process of low beta plasmas, the enhanced oblique daughter waves force the perpendicular ion heating ion heating.

Viñas and Goldstein 1991(a,b) discovered indeed new channels of parametric instabilities when the dimension of the analyzed MHD system is increased from 1-D to 2-D thus allowing that the daughter and side-band modes to obliquely propagate with respect to the field-aligned Alfvén pump wave. Some of these new instabilities, e.g., the filamentation instability, have been identified in later

**Fig. A.3.** Top: Comparison between obliquely-propagating daughter modes  $\delta B(-8,8)$  in the 3-D setup (left) and 2-D setup (right) versus the elapsed time. The field-aligned Alfvén pump mode  $\delta B(10,0)$  and its corresponding Alfvén daughter mode  $\delta B(-8,0)$  are also given as reference. Bottom: Time evolution of the perpendicular projection  $\delta \rho$  (0,8) of the obliquely-propagating density mode  $\delta \rho$  (18,8) coupled to the Alfven mode  $\delta B(-8,8)$ . The field-aligned density modes  $\delta \rho$  (18,0) are also plotted for both 3-D (left) and 2-D (right) setups.

- 215 MHD simulations (Ghosh et al. 1993, 1994) by comparison with the analytical predictions of Viñas and Goldstein 1991(a,b). Therefore, a more exact answer to the referee's question implies an analytical study as that one of Viñas and Goldstein 1991(ab) by considering the azimuthal propagation angles of daughter modes. Naively speaking, we would not expect significant changes or dependencies of the parametric instabilities with respect to the azimuthal angle around the mean magnetic
- 220 field. Analytic treatments concerning oblique instabilities developed by parallel propagating Alfven pump waves have no longer continued since Viñas and Goldstein 1991(a,b) papers.

Changes in the manuscript: Page:10, Line:15-30

- 225 "On the other hand, the 3D case opens new channels of the parametric instabilities in the sense that there is a larger population of obliquely propagating waves due to a degree of freedom in the azimuthal direction around the mean magnetic field. As the oblique propagating waves play an important role in the heating process of low beta plasmas, the enhanced oblique daughter waves lead to a more efficient ion heating in the perpendicular direction.
- 230 Viñas and Goldstein 1991(a,b) discovered new channels of parametric instabilities when the dimension of the analyzed MHD system is increased from 1-D to 2-D thus allowing that the daughter and side-band modes to obliquely propagate with respect to the field-aligned Alfvén pump wave. Some of these new instabilities, e.g., the filamentation instability, have been identified in later MHD simulations (Ghosh et al. 1993, 1994) by comparison with the analytical predictions of Viñas and
- 235 Goldstein 1991(a,b). To emphasize new instabilities particularly enhanced in 3D with respect to 2D setups an analytical study is needed by considering both the oblique and azimuthal propagation angles of daughter modes. However, analytic treatments concerning oblique instabilities developed by parallel propagating Alfven pump waves have no longer continued since Viñas and Goldstein 1991(a,b) papers. An outcome of these former theoretical studies confirmed in the present numer-
- 240 ical study is that the obliquely-propagating daughter waves are stronger and play a more important

---

## Author Response (AR4)

**Reply to referee comments**

On heating of solar wind protons by the parametric decay of large amplitude
Alfvén waves

Manuscript ID: angeo-2018-14 (AngeoComm)

5 H. Comişel, Y. Nariyuki, Y. Narita, and U. Motschmann

We thank reviewer for evaluating our paper.

*Reviewer:*
*The last minor remark is that the following text belongs rather to Discussion than to the description*
*of the model: "The used spatial resolution for the field ... Thus we conclude that the numerical*
10 *heating does not play a significant role compared to the physical heating."*

==================================================================

Reply:
We follow the reviewer's remark.

15

Changes in the manuscript:
The mentioned text is moved from section 2 to section 4.
Page:10, Line:11 to Page:11, Line:2
"The used spatial resolution for the field quantities (magnetic field, electric field, and velocity mo-
20 ments) is close to the ion inertial length and the proton gyroradius ($\rho_i \sim 0.1 d_i$) or smaller spatial
gradients cannot be resolved. The magnetic field within a numerical cell is overall homogeneous
with the linear interpolations at the particle position between mesh points or due to the wave mag-
netic field. Thus, the perpendicular projection of the proton motion is nearly a circle and this circular
gyration is resolved by about 100 time steps. Gradients become important over about 10 gyroradii
25 and not just over one gyration. We are warned that numerical heating could have some contribu-
tion in our simulations. Among various candidate mechanisms causing numerical heating one may
specify: the numerical noise given by the statistical representation of the distribution functions,
the rounding error or cutoff error when evaluating the differential operator, the absorption of the
numerically-arising electric (possibly the electrostatic field) by the ions, and the random scattering
30 due to the numerically fluctuating magnetic field (here the magnetic diffusion may be applicable).
The numerical free energy occurring in the system can be converted in wave energy. This wave
energy can be absorbed by particles and heating of the plasma. The heating effects described above
can be compensated by using a suitable resistivity parameter, a smoothing procedure for the mag-
netic field, and numerical tests including various parameters. We have tested simulation runs with or
35 without using pump wave by varying the number of particles per cell, time steps $\delta t$, and grid sizes to
find out sufficient energy accuracy (within 5% for 500 elapsed ion-gyroperiods). Thus we conclude
that the numerical heating does not play a significant role compared to the physical heating."